# The oxidized phospholipid oxPAPC protects from septic shock by targeting the non-canonical inflammasome in macrophages

Lan H. Chu[1,2], Mohanalaxmi Indramohan[1], Rojo A. Ratsimandresy[1], Anu Gangopadhyay[1,2], Emily P. Morris[1], Denise M. Monack[3], Andrea Dorfleutner[1] & Christian Stehlik[1,4]

Lipopolysaccharide (LPS) of Gram-negative bacteria can elicit a strong immune response. Although extracellular LPS is sensed by TLR4 at the cell surface and triggers a transcriptional response, cytosolic LPS binds and activates non-canonical inflammasome caspases, resulting in pyroptotic cell death, as well as canonical NLRP3 inflammasome-dependent cytokine release. Contrary to the highly regulated multiprotein platform required for caspase-1 activation in the canonical inflammasomes, the non-canonical mouse caspase-11 and the orthologous human caspase-4 function simultaneously as innate sensors and effectors, and their regulation is unclear. Here we show that the oxidized phospholipid 1-palmitoyl-2-arachidonoyl-*sn*-glycero-3-phosphorylcholine (oxPAPC) inhibits the non-canonical inflammasome in macrophages, but not in dendritic cells. Aside from a TLR4 antagonistic role, oxPAPC binds directly to caspase-4 and caspase-11, competes with LPS binding, and consequently inhibits LPS-induced pyroptosis, IL-1β release and septic shock. Therefore, oxPAPC and its derivatives might provide a basis for therapies that target non-canonical inflammasomes during Gram-negative bacterial sepsis.

---

[1] Division of Rheumatology, Department of Medicine, Feinberg School of Medicine, Northwestern University, Chicago, Illinois 60611, USA. [2] Driskill Graduate Program in Life Sciences, Feinberg School of Medicine, Northwestern University, Chicago, Illinois 60611, USA. [3] Department of Microbiology and Immunology, Stanford School of Medicine, Stanford University, Stanford, Stanford, California 94305, USA. [4] Robert H. Lurie Comprehensive Cancer Center, Interdepartmental Immunobiology Center and Skin Disease Research Center, Feinberg School of Medicine, Northwestern University, Chicago, Illinois 60611, USA. Correspondence and requests for materials should be addressed to A.D. (email: a-dorfleutner@northwestern.edu) or to C.S. (email: c-stehlik@northwestern.edu)

Innate immune pattern recognition receptors are essential for sensing conserved self and non-self danger signals to initiate inflammatory responses that eliminate pathogens and initiate wound repair. Lipopolysaccharide (LPS) consisting of lipid A, core sugars, and O-antigen constitutes the major component in the outer membrane of Gram-negative bacteria and is a potent innate immune stimulator[1]. Although extracellular LPS is chaperoned by CD14 from the LPS-binding protein to the TLR4/MD-2 transmembrane LPS receptor complex at the cell surface and induces transcription of proinflammatory genes[2,3], intracellular LPS delivered by Gram-negative bacteria escaping the phagosome[4–6] or by extracellular bacterial outer membrane vesicles (OMVs)[7] activates the non-canonical inflammasome caspases[8,9]. Whereas caspase-1 is activated by the highly regulated canonical inflammasome complex[10–12], caspase-11 and the human orthologs caspase-4 and caspase-5 lack an upstream sensory complex, and are activated by directly binding to LPS through the N-terminal caspase recruitment domain (CARD)[13]. This activation results in pyroptotic cell death, cytokine release by the canonical inflammasome, and consequently reduced survival in response to LPS-induced shock in a TLR4-independent manner[8,9,14]. Hence, non-canonical inflammasome caspases function directly as sensors and effectors that proteolytically activate gasdermin D (GSDMD), which mediates pyroptosis[15,16]. Notably, the negative regulatory mechanisms of non-canonical caspase activation are still mostly unknown.

Naturally occurring phospholipids, such as 1-palmitoyl-2-arachidonoyl-sn-glycero-3-phosphorylcholine (PAPC), are located in cell membranes and lipoproteins. The oxidation product of PAPC, oxidized PAPC (oxPAPC), is a bioactive principal component of minimally modified low-density lipoproteins and modulates inflammatory responses[17,18]. However, the precise function of oxPAPC is controversial and probably context dependent, as both proinflammatory and anti-inflammatory effects have been described[18]. oxPAPC functions as a TLR4 agonist to induce expression of interleukin (IL)-6, IL-8, and MCP-1, and promotes adhesion of monocytes[19,20]. However, oxPAPC has also TLR4 antagonistic activity and interferes with TLR4-mediated LPS responses[21–24]. Importantly, much lower oxPAPC concentrations are required to antagonize TLR4 activation than those required for proinflammatory responses[23]. Consequently, oxPAPC has a potent anti-inflammatory activity and prevents tissue damage in response to LPS-induced and TLR4-mediated endotoxemia[25]. High concentrations of oxPAPC have been proposed to have a weak agonistic activity for caspase-11 in dendritic cells (DCs), but not macrophages, and promote adaptive immune responses[26]. Even higher concentrations of the individual oxPAPC components 1-palmitoyl-2-(5'-oxo-valeroyl)-sn-glycero-3-phosphocholine (POVPC) and 1-palmitoyl-2-glutaryl-sn-glycero-3-phosphocholine (PGPC) have been suggested to activate caspase-11 and the canonical NLRP3 inflammasome in macrophages[27,28]. However, this finding is in contrast to earlier reports, demonstrating that oxPAPC prevents DC activation, maturation, and cytokine release, and prevents sepsis[25,29]. Hence, the function of oxPAPC during inflammation is still unclear.

Here we report an anti-inflammatory function of oxPAPC in macrophages, which directly binds to caspase-4 and caspase-11, prevents caspase interaction with LPS and thereby inhibits LPS-mediated pyroptosis, cytokine release, and septic shock.

## Results

### oxPAPC inhibits LPS-induced pyroptosis in macrophages.
As the role of oxPAPC in the LPS response is controversial, we first tested the effect on LPS-induced IL-6 production, which is dependent on TLR4. As expected, activation of bone marrow-derived macrophages (BMDMs) (Supplementary Fig. 1a and 2) with LPS-induced IL-6 release only in wild type (WT), but not $Tlr4^{-/-}$ BMDMs, whereas oxPAPC alone did not induce IL-6 release at any tested concentration (Fig. 1a). Simultaneous LPS and oxPAPC treatment showed reduced IL-6 release compared with LPS treatment alone, and this antagonistic effect on TLR4 signaling was further enhanced if BMDMs were pre-incubated with oxPAPC before LPS stimulation (Fig. 1a). In contrast, oxPAPC failed to inhibit IL-6 release when we treated BMDMs first with LPS, followed by the addition of oxPAPC (Fig. 1a), indicating that oxPAPC elicits an inhibitory response to LPS-mediated TLR4 activation in macrophages, even when using 10-fold higher oxPAPC concentrations (data now shown). Besides TLR4[30], also the non-canonical inflammasome-linked caspases directly bind the lipid A moiety of LPS via their CARD[13]. We therefore sought to determine, whether oxPAPC can affect caspase-11 activity in macrophages, as it has been shown to weakly activate caspase-11-mediated IL-1β release in DCs[26]. Although caspase-1 is responsible for the maturation and release of the proinflammatory cytokines IL-1β and IL-18, and initiating pyroptosis in response to multiple Gram-negative bacteria, caspase-11 initiates pyroptosis in response to cytosolic LPS to eliminate the intracellular replication niche for Gram-negative bacteria[4–6]. To test whether oxPAPC affects LPS-induced pyroptosis, we transfected LPS with or without oxPAPC into BMDMs and measured the release of lactate dehydrogenase (LDH) into culture supernatants. To upregulate caspase-11 expression, we primed BMDMs with the TLR2 agonist Pam3CSK4 before LPS transfection[8,9,31]. Contrary to the modest 25% inhibition of TLR4, co-transfection of oxPAPC with LPS completely abrogated LDH release, whereas PAPC had no significant effect and the minor reduction may be attributed to spontaneous PAPC oxidation. Neither PAPC nor oxPAPC alone had any effect on pyroptosis (Fig. 1b). Significantly, to completely block pyroptosis only a 2-fold excess of oxPAPC over LPS was needed (Fig. 1b), but even a 20-fold excess of oxPAPC could not efficiently block TLR4 (Fig. 1a). oxPAPC may affect TLR2 signaling, and we therefore primed cells with the TLR3 agonist poly(I:C) that is not affected by oxPAPC[21]. TLR2 or TLR3 priming showed no differences in the ability of oxPAPC to prevent LPS-induced pyroptosis (Fig. 1c). Furthermore, simultaneous transfection of oxPAPC and LPS, as well as LPS transfection after 30 min preincubation with oxPAPC in WT and $Tlr3^{-/-}$ BMDMs, and peritoneal macrophages (PMs), completely blocked LDH release (Fig. 1d and Supplementary Fig. 1b), demonstrating that the oxPAPC inhibition of LPS-induced pyroptosis does not require oxPAPC transfection. $Casp11^{-/-}$ and poly(I:C)-primed $Tlr3^{-/-}$ BMDMs and PMs were resistant to LPS-induced pyroptosis. However, oxPAPC reduced LPS-induced pyroptosis in Pam3CSK4-primed $Tlr3^{-/-}$ BMDMs and PMs, indicating that oxPAPC is able to block LPS-induced pyroptosis regardless of priming (Fig. 1d and Supplementary Fig. 1b). Furthermore, neither oxPAPC treatment nor transfection in the absence of LPS resulted in increased toxicity (Supplementary Fig. 1c and 3). As BMDMs and PMs showed a comparable response, we performed most experiments with poly(I:C)-primed BMDMs. Although the scavenger receptor CD36 has been implicated in signaling and internalization of oxidized phospholipids[32], it was dispensable for oxPAPC-mediated inhibition of pyroptosis (Supplementary Fig. 1d). Comparable intracellular oxPAPC levels were also detected in $Tlr4^{-/-}$ and WT BMDMs, indicating that also TLR4 was dispensable for the oxPAPC effect (Supplementary Fig. 1e, 4). To further demonstrate specificity of this response, we titrated oxPAPC from 10 to 500 ng, which resulted in a dose-dependent inhibition of pyroptosis (Fig. 1e). Notably, we achieved 50% reduction in LDH release with only a

0.5-fold excess of oxPAPC over LPS, which is 40-fold less than required for blocking TLR4 signaling by 50%, indicating that oxPAPC more potently inhibited pyroptosis than TLR4 signaling. Peroxidation of PAPC results in a heterogeneous mixture of bioactive phospholipids, including PGPC and POVPC. To account for this heterogeneous mixture, we used three different oxPAPC preparations, which all demonstrated a very comparable inhibition of LPS-induced pyroptosis (Supplementary Fig. 1f–h).

*Casp11*[−/−] BMDMs did not show any cytotoxicity in response to LPS transfection and co-transfection of oxPAPC had no effect in these cells (Fig. 1f and Supplementary Fig. 1i). As oxPAPC affects TLR4 signaling[21,25], we transfected LPS and LPS/oxPAPC into *Tlr4*[−/−] BMDMs to ensure that the inhibitory effect of oxPAPC was not mediated by TLR4. Pyroptosis and oxPAPC-mediated inhibition was comparable in *Tlr4*[−/−] and WT BMDMs (Fig. 1g), excluding TLR4-based oxPAPC effects. Although sequential LPS

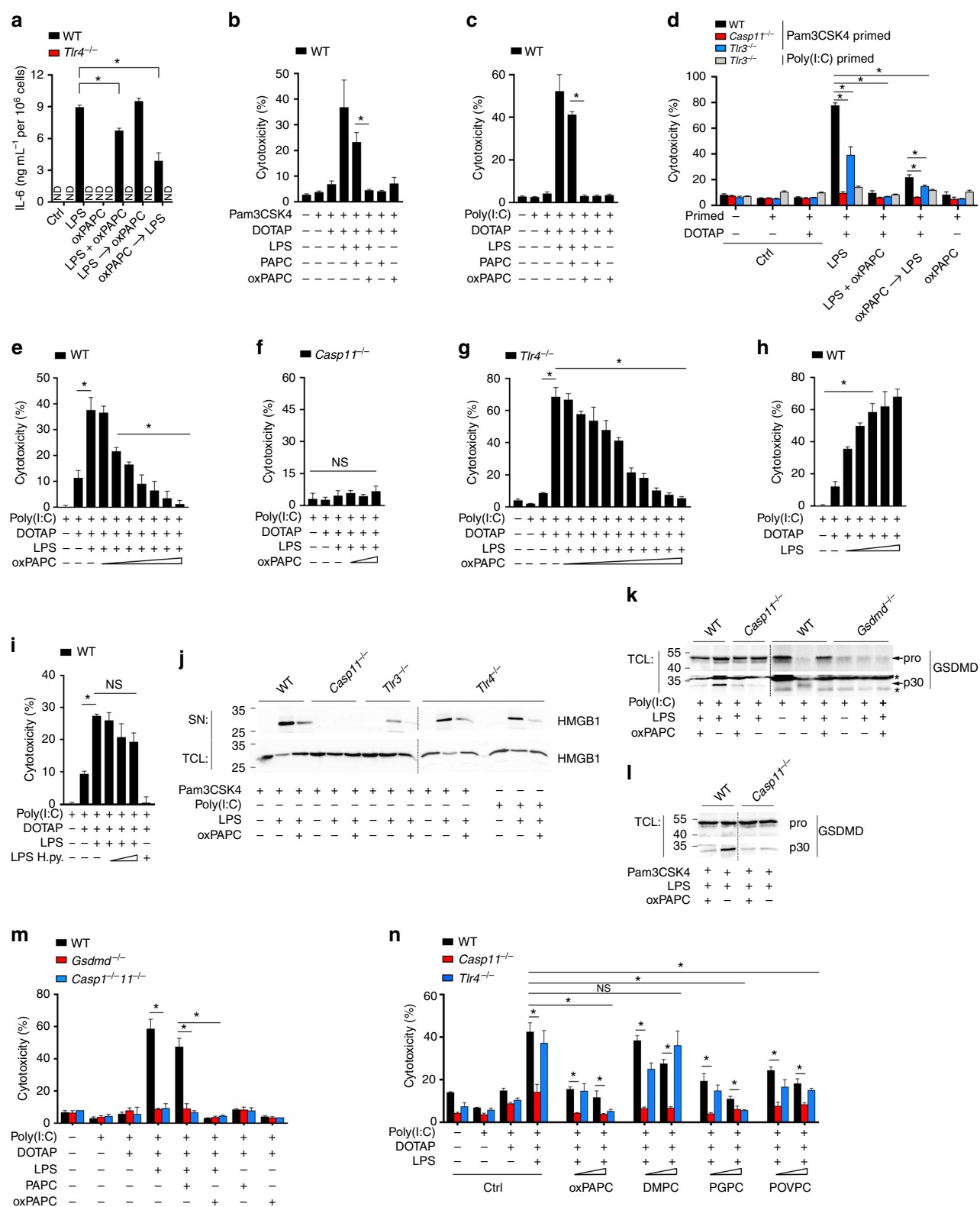

and oxPAPC transfection indicated a specific oxPAPC response and ruled out a possible saturation of the transfection reagent N-[1-(2,3-Dioleoyloxy)propyl]-N,N,N-trimethylammonium methyl-sulfate (DOTAP) in LPS/oxPAPC co-transfection experiments, we titrated LPS starting from the amount used in above experiments. Indeed, increased LPS transfection augmented LDH release (Fig. 1h), indicating that DOTAP was not saturated under our experimental conditions. We also directly eliminated any potential oxPAPC-mediated impaired LPS uptake, as intracellular flow cytometry revealed identical LPS staining in the presence and absence of oxPAPC (Supplementary Fig. 1j and 5). Electropora-tion further excluded liposome-based effects (Supplementary Fig. 1k, l). In contrast to *Escherichia coli* hexa-acylated lipid A, the tetra-acylated lipid A from *Helicobacter pylori* does neither sti-mulate TLR4 nor caspase-11[5,8]. We therefore attempted to out-compete *E. coli* LPS with excess of *H. pylori* LPS, which neither induced pyroptosis nor reduced *E. coli* LPS-induced pyroptosis (Fig. 1i). The danger signal high-mobility group 1 (HMGB1) is released by pyroptosis and, accordingly, HMGB1 levels in the culture supernatant of LPS-transfected WT BMDMs were reduced, when we co-transfected LPS/oxPAPC and was lost in *Casp11*[−/−], but not in *Tlr3*[−/−] and *Tlr4*[−/−] BMDMs (Fig. 1j). GSDMD is the downstream inflammatory caspase substrate[15,33], and the cleaved N-terminal fragment promotes pyroptosis through lytic plasma membrane pores[34–38]. Accordingly, trans-fection of LPS into either poly(I:C)- (Fig. 1k) or Pam3CSK4-primed BMDMs (Fig. 1l) resulted in the appearance of the cleaved GSDMD p30, which was completely prevented in *Casp11*[−/−] and in *Gsdmd*[−/−] BMDMs, and in cells co-transfected with LPS and oxPAPC (Fig. 1k, l). LPS-induced pyroptosis was also abolished in *Gsdmd*[−/−] and *Casp1*[−/−]*Casp11*[−/−] BMDMs or oxPAPC-treated WT BMDMs (Fig. 1m). Peroxidation of PAPC yields a number of structurally distinct phospholipids and the major PAPC perox-idation products POVPC and PGPC inhibited LPS-induced pyr-optosis comparable to oxPAPC, whereas the non-oxidized 1,2-dimyristoyl-sn-glycero-3-phosphocholine (DMPC) did not affect pyroptosis (Fig. 1n and Supplementary Fig. 6). Our results demonstrate that cytosolic oxPAPC inhibits LPS-induced pyr-optosis in macrophages.

**oxPAPC inhibits IL-1β release by macrophages.** Caspase-11-mediated IL-1β release depends on the non-canonical activation of the NLRP3 inflammasome[39]. Although POVPC has been proposed as an NLRP3 ligand[28], neither PAPC nor oxPAPC alone promoted IL-1β release in BMDMs (Fig. 2a). However, transfection of LPS caused IL-1β release in primed BMDMs, which required caspase-11 and the canonical NLRP3 inflamma-some, including the adaptor apoptosis-associated speck-like protein containing a CARD (ASC), as *Nlrp3*[−/−], *Asc*[−/−], *Casp11*[−/−] and *Casp1*[−/−]*Casp11*[−/−] BMDMs, as well as oxPAPC, but not

PAPC, treated BMDMs displayed impaired IL-1β release (Fig. 2b). In addition to NLRP3, likely another ASC-dependent sensor contributes to this response, as *Nlrp3*[−/−], but not *Asc*[−/−] and *Casp1*[−/−]*Casp11*[−/−] BMDMs showed residual IL-1β release. In contrast to pyroptosis, which proceeded independent of TLR4, IL-1β release was partially dependent on TLR4 (Fig. 2b), sug-gesting that even liposomal delivery of LPS engaged TLR4. Indeed, transcription of *Il1b* was comparably reduced in *Tlr4*[−/−] BMDMs at 1 and 2 h, and pro-IL-1β protein at 4 h after LPS transfection (Supplementary Fig. 7). IL-1α is co-secreted with IL-1β and is partially dependent on inflammasome activation[40]. Cytosolic LPS-induced IL-1α release was caspase-11- and par-tially canonical NLRP3 inflammasome- and TLR4-dependent (Fig. 2c). In contrast, cytosolic LPS-induced IL-6 secretion was completely independent of caspase-11 and the canonical NLRP3 inflammasome, but was partially dependent on TLR4 (Fig. 2d). Neither PAPC nor oxPAPC alone induced IL-6 release and PAPC, and, more importantly, also oxPAPC did not inhibit cytosolic LPS-induced IL-6 secretion. Furthermore, as observed for pyroptosis, IL-1β and IL-1α release required GSDMD (Fig. 2e, f). Hence, our results demonstrate that cytosolic oxPAPC also inhibits cytosolic LPS-induced IL-1β release in macrophages.

As oxPAPC has been proposed as a weak caspase-11 agonist in DCs[26], we isolated DCs from WT, *Casp11*[−/−], *Casp1*[−/−], *Asc*[−/−], and *Tlr4*[−/−] BM to determine the contribution of the canonical and non-canonical inflammasome and TLR4. Highly pure FLT3L/granulocyte–macrophage colony-stimulating factor (GM-CSF)-differentiated BM-derived DCs (BMDCs) (Supplementary Fig. 8a and 9) were more responsive to LPS compared with BMDMs, as LPS priming alone already caused some IL-1β release, which was independent of caspase-11, but dependent on TLR4 and the canonical NLRP3 inflammasome (Fig. 3a, b). Addition of oxPAPC to LPS-primed BMDCs did not show agonistic activity, even when using oxPAPC concentrations 25 times more than required to fully block caspase-11 in BMDMs. Only when we treated BMDCs with 100 and 200 μg mL⁻¹, we observed a very weak agonistic activity, compared with the canonical NLRP3 activator nigericin (Fig. 3a). Although DCs showed more basal LDH release than BMDMs, none of the treatments further increased LDH release, which was indepen-dent of TLR4, the canonical and non-canonical inflammasome (Supplementary Fig. 8b, c). We confirmed release of mature IL-1β (p17) by immunoblot (Supplementary Fig. 8d). FLT3L/GM-CSF differentiation yields a more homogeneous BMDC population, but can display functional differences compared with GM-CSF-differentiated BMDCs[41], but even highly pure GM-CSF-differentiated BMDCs (Supplementary Fig. 8e, 10) showed comparable results (Supplementary Fig. 8f). However, even 200 μg mL⁻¹ did not promote any IL-1β release in LPS-primed BMDMs and, contrary to BMDCs, LPS priming alone did not promote IL-1β release (Fig. 3a). oxPAPC-induced IL-1β release

**Fig. 1** oxPAPC inhibits LPS-induced pyroptosis by targeting caspase-11 in macrophages. **a** WT and *Tlr4*[−/−] BMDMs were treated with LPS (100 ng mL⁻¹) and oxPAPC (2 μg mL⁻¹) for 6 h or pre-incubated with oxPAPC or LPS for 0.5 h as indicated, and IL-6 was determined in culture supernatants by ELISA. **b, c, e–i, m** (**b, c, e–l, m**) WT BMDMs were primed with Pam3CSK4 (1 μg mL⁻¹) or poly(I:C) (1 μg mL⁻¹) for 6 h and then transfected with **b, c, e–g, m** LPS (50 ng per well) and **b, c, m** oxPAPC or PAPC (100 ng per well), **e** oxPAPC (10, 30, 50, 75, 100, 250, 500 ng per well), **f** oxPAPC (10, 50, 1875 ng per well), **g** oxPAPC (10, 20, 30, 40, 50, 75, 100, 150, 200, 500 ng per well), **h** LPS (50, 75, 100, 150, 200 ng per well), and **i** *E. coli* LPS (50 ng per well) and *H. pylori* (H.py) LPS (25, 100, 500 ng per well) as indicated. LDH release after 2 h was determined and presented as % cytotoxicity compared to maximum LDH release. **d** BMDMs of the indicated genotypes were primed with Pam3CSK4 (1 μg mL⁻¹) or poly(I:C) (1 μg mL⁻¹) for 6 h and then transfected with LPS (50 ng per well) and either co-transfected with oxPAPC (100 ng per well) or pre-treated with oxPAPC (2 μg per well, indicated by an arrow) and LDH release determined as above. **j–l** BMDMs of the indicated genotype were primed as above and transfected with LPS (50 ng per well) and oxPAPC (100 ng per well) as indicated and culture supernatants (SN) or total cell lysates (TCL) were analyzed by immunoblot after 2 h or 3 h for **j** HMGB1 and **k, l** GSDMD, showing the pro-form (pro) and cleaved (p30) GSDMD. **n** WT and *Casp1*[−/−] BMDMs were primed with poly(I:C) (1 μg mL⁻¹) for 6 h and then transfected with LPS (50 ng per well), and either co-transfected with oxPAPC, PGPC, POVPC, and DMPC (50 and 100 ng per well) and LDH release determined as above. Data are representative of at least three independent experiments of at least triplicate samples. Error bars indicate ± SD; * $P < 0.05$ by two-tailed unpaired $t$-test

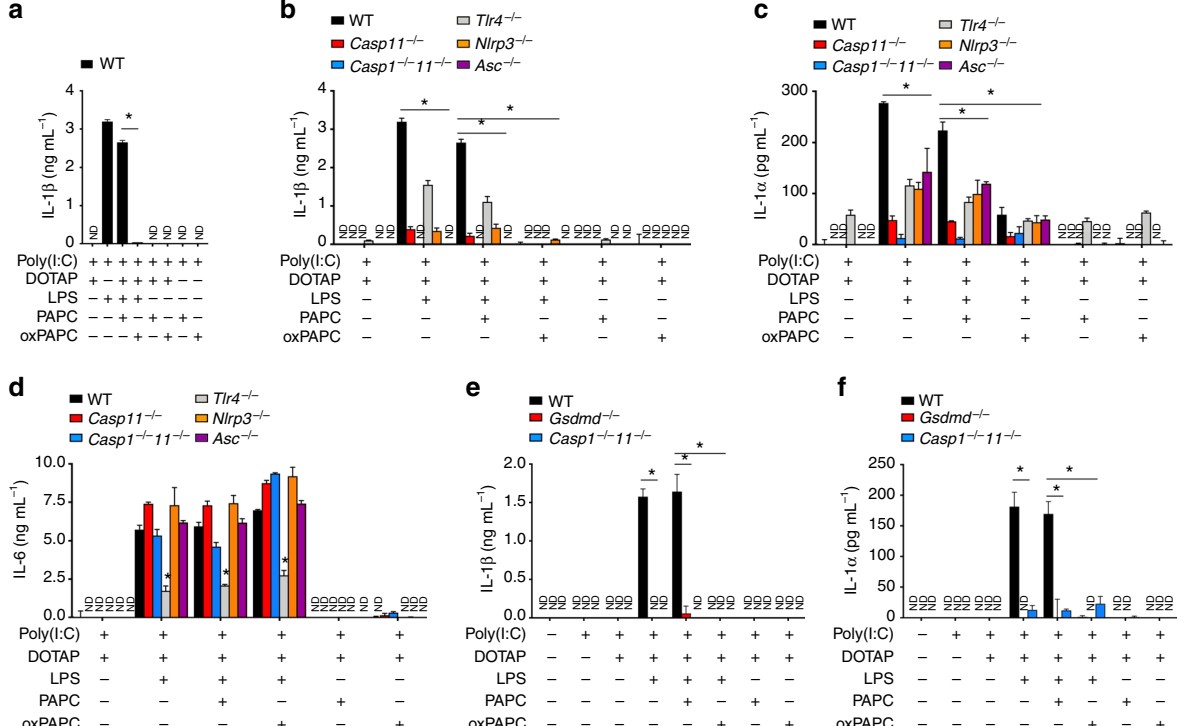

**Fig. 2** oxPAPC inhibits cytosolic LPS-induced cytokine release in macrophages. **a–f** BMDMs of the indicated genotypes were primed with poly(I:C) (1 μg mL$^{-1}$) for 6 h and then transfected with LPS (50 ng per well) and oxPAPC or PAPC (100 ng per well), and culture supernatants analyzed for secreted **a**, **b**, **e** IL-1β, **c**, **f** IL-1α, and **d** IL-6 by ELISA after 4 h. Data are representative of at least three independent experiments of at least triplicate samples. Error bars indicate ± SD; *P < 0.05 by two-tailed unpaired t-test

did not specifically require TLR4-mediated priming, as we observed an identical response following TLR2 priming with Pam3CSK4 (Fig. 3c). As observed above for BMDMs, also *Tlr4*$^{-/-}$ BMDCs revealed slightly reduced IL-1β release, even when primed with Pam3CSK4 (Fig. 3c). Importantly, this weak, high-dose LPS/oxPAPC-mediated IL-1β release was not at all reduced in *Casp11*$^{-/-}$ BMDCs, but was lost in *Tlr4*$^{-/-}$, *Asc*$^{-/-}$, *Casp1*$^{-/-}$, and *Nlrp3*$^{-/-}$ BMDCs, indicating that caspase-11 is not required for LPS/oxPAPC-induced IL-1β release in BMDCs (Fig. 3a–c and Supplementary Fig. 8f). Importantly, oxPAPC was not able to promote IL-1β release in naive BMDCs, indicating that a two-step mechanism is required. We propose that the apparent cell-type-specific response to oxPAPC is based on differences in oxPAPC uptake, which favors extracellular or intracellular recognition of oxPAPC. We find that oxPAPC uptake is at least 10-fold more efficient in BMDMs than in BMDCs (Fig. 3d and Supplementary Fig. 11), resulting in primarily TLR4-mediated cell surface responses in BMDCs. Upon transfection of LPS and oxPAPC into the cytosol of LPS-primed BMDCs, the LPS-TLR4 cell surface response is re-directed to a cytosolic caspase-11-dependent release of IL-1β, which is sensitive to oxPAPC inhibition, although TLR4 is still required for LPS priming (Fig. 3e). Hence, the inability of BMDCs to take up oxPAPC into the cytosol dictates a TLR4-driven response.

**oxPAPC inhibits bacterial activation of caspase-11.** We next determined the response of oxPAPC following pathophysiological-relevant delivery of LPS. Extracellular bacteria deliver LPS to the cytosol of macrophages and activate the non-canonical inflammasome through OMVs[7]. *E. coli* OMVs caused pyroptosis in WT, but not in *Casp11*$^{-/-}$ BMDMs, and pre-treatment with oxPAPC completely prevented pyroptosis

(Fig. 4a). Accordingly, also IL-1β release after OMV treatment was dependent on caspase-11 in BMDMs and was also prevented by pre-treatment with oxPAPC (Fig. 4b). Caspase-11 senses LPS from intracellular bacteria escaping the vacuole[4,9]. *Salmonella typhimurium* uses two distinct type III secretion systems (T3SS) encoded by SPI-1 and SPI-2, which translocate different effectors into the cytosol[42]. The canonical NLRC4 inflammasome ligand flagellin translocates into the cytosol through SPI-1 and the vacuole stabilizing SifA through SPI-2 and distinct growth conditions favor SPI-1 or SPI-2 expression. *S. typhimurium* growing in the logarithmic phase express SPI-1 and flagellin, which are detected by NLRC4[43]. In the stationary phase, which mimics the vacuolar environment, *S. typhimurium* expresses SPI-2 and represses SPI-1 and flagellin to evade NLRC4 detection to establish an intracellular niche[44,45]. However, deletion of *sifA* causes vacuolar rupture, bacterial escape to the cytosol, and detection by caspase-11[4,5]. Hence, for caspase-11 activation *S. typhimurium* Δ*sifA* was grown to the stationary phase before infection, which triggered caspase-11-dependent pyroptosis that was abrogated by pre-transfection or pre-treatment with oxPAPC, but not PAPC (Fig. 4c). We obtained comparable results with *S. typhimurium* Δ*sifA/flgB*, ΔSPI-1/Δ*fla*, and ΔSPI-1/ΔSPI-2 double mutants (Fig. 4d). However, oxPAPC was not able to prevent NLRC4 inflammasome-dependent pyroptosis (Fig. 4e) and IL-1β release (Fig. 4f) triggered by flagellin, which further demonstrates that oxPAPC specifically inhibits the non-canonical inflammasome.

**oxPAPC inhibits caspase-4 in human macrophages.** Human caspase-4 and caspase-5 are implicated in the cytosolic LPS response[13]. However, silencing of caspase-4 was sufficient to impair cytosolic LPS-induced pyroptosis[46–48]. We transfected

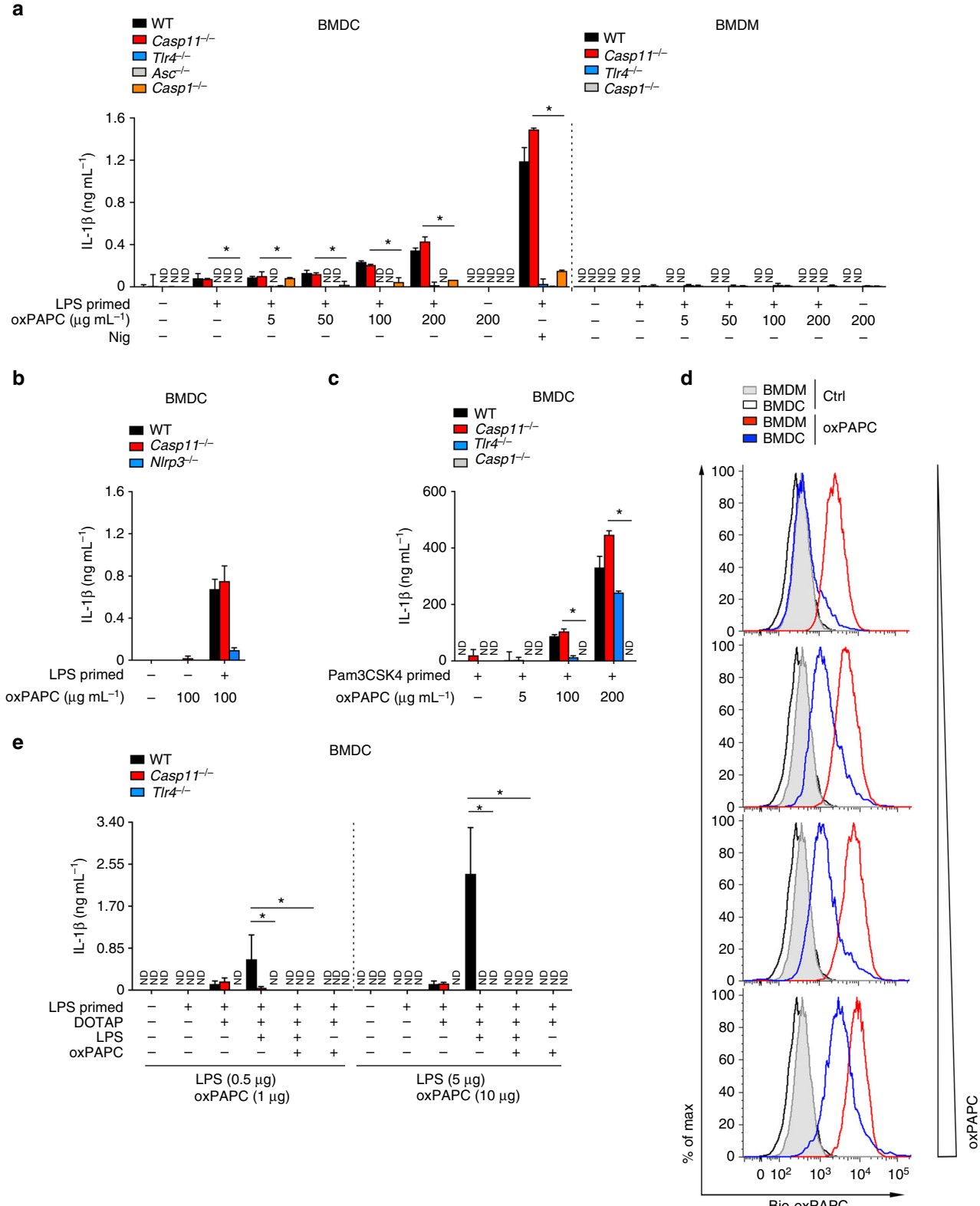

**Fig. 3** oxPAPC does not inhibit LPS and caspase-11-induced cytokine release in DCs. **a–c** FLT3L/GM-CSF-differentiated BMDCs or M-CSF-differentiated BMDMs of the indicated genotypes were primed with **a**, **b** LPS (O55:B5, 1 µg mL$^{-1}$) or **c** Pam3CSK4 (1 µg mL$^{-1}$) for 3 h and then treated with oxPAPC (0 to 200 µg mL$^{-1}$) or only oxPAPC (0, 100, or 200 µg mL$^{-1}$) as indicated and culture supernatants analyzed for secreted IL-1β by ELISA 18 h after LPS treatment. **d** BMDMs and BMDCs were treated with biotinylated oxPAPC (1, 2, 5, and 10 µg mL$^{-1}$) (Bio-oxPAPC) for 30 min and intracellular oxPAPC quantified by flow cytometry. **e** FLT3L/GM-CSF-differentiated BMDCs of the indicated genotypes were primed with LPS (O55:B5, 1 µg mL$^{-1}$) for 3 h and then transfected with LPS (0.5 or 5 µg per well) and oxPAPC (1 or 10 µg per well) as indicated and culture supernatants analyzed for secreted IL-1β after 4 h. Data are representative of at least three independent experiments of at least triplicate samples. Error bars indicate ± SD; *P < 0.05 by two-tailed unpaired t-test

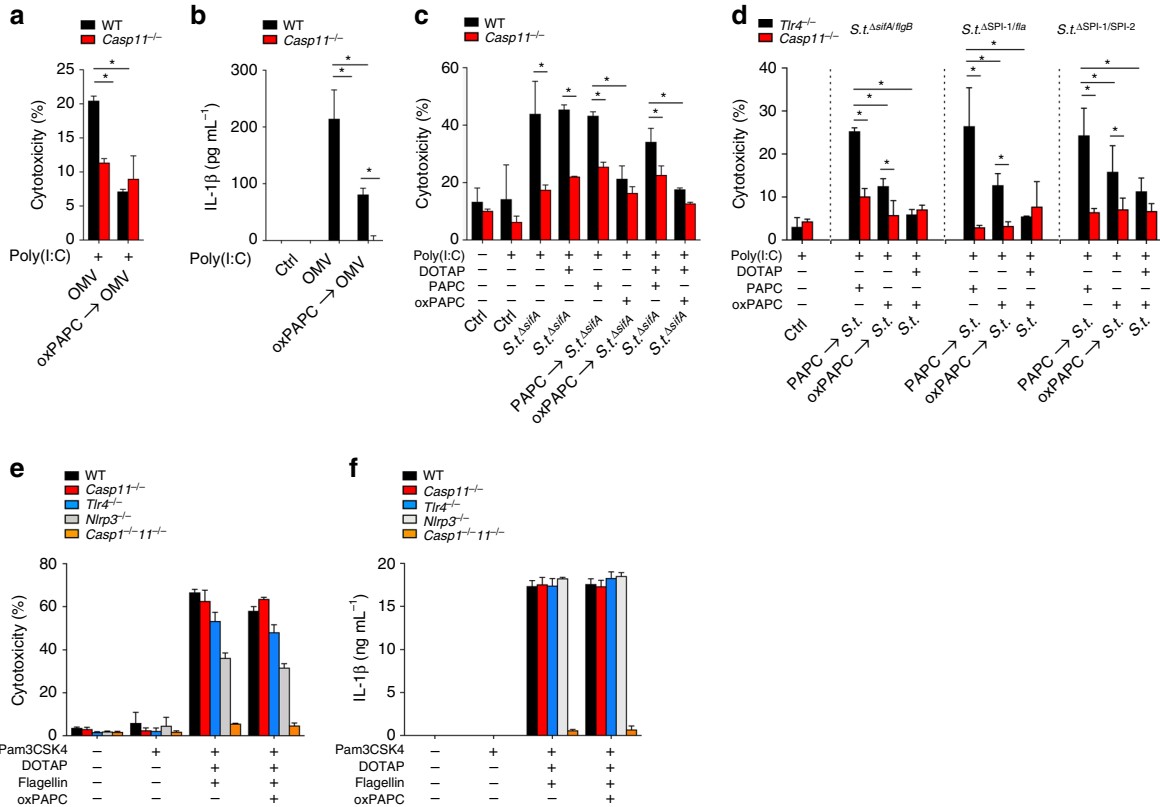

**Fig. 4** oxPAPC inhibits pyroptosis and cytokine release during Gram-negative bacterial infection. **a**, **b** WT or *Casp11*⁻/⁻ BMDM were either pre-treated with PBS or oxPAPC (1 μg per well), followed by incubation with OMVs (20 μg) for 4 h. **a** LDH release after 4 h was determined and presented as % cytotoxicity compared with maximum LDH release. **b** IL-1β release was determined after 4 h by ELISA. **c**, **d** WT or *Casp11*⁻/⁻ BMDM were either untreated, pre-treated with PAPC or oxPAPC (1 μg per well, indicated by an arrow), or transfected with oxPAPC (100 ng per well), followed by infection with *S. typhimurium* (*S.t.*) Δ*sifA*, Δ*sifA*/Δ*flgB*, ΔSPI-1/Δ*fla*, and ΔSPI-1/ΔSPI-2, as indicated, and cytotoxicity determined as above. **e**, **f** BMDM of the indicated genotypes were transfected with *S. typhimurium* flagellin (15 ng per well) in the presence or absence of oxPAPC (100 ng per well), and cytotoxicity and IL-1β release determined as above. Data are representative of at least three independent experiments of at least triplicate samples. Error bars indicate ± SD; *P < 0.05 by two-tailed unpaired *t*-test

LPS or co-transfected LPS/oxPAPC into poly(I:C)-primed human THP-1 cells, and oxPAPC completely inhibited cytosolic LPS-induced LDH release (Fig. 5a), HMGB1 release, and GSDMD cleavage (Fig. 5b). LPS-induced pyroptosis was also lost in CASP4 and GSDMD small interfering RNA (siRNA)-transfected cells compared with control siRNA (Fig. 5c). Further, pre-treatment with oxPAPC, but not PAPC, prevented cytosolic LPS-induced pyroptosis (Fig. 5d). We controlled siRNA-mediated knockdown of caspase-4 and GSDMD by immunoblot and found very efficient knockdown for caspase-4 but less efficient knockdown for GSDMD (Fig. 5e), which explained the partial response observed in GSDMD-silenced cells. Although oxPAPC, but not PAPC, prevented cytosolic LPS-induced pyroptosis in human cells, it did not require the canonical NLRP3 inflammasome, as CRISPR-mediated knockout or short hairpin RNA (shRNA)-mediated silencing of *NLRP3*, ASC, and *CASP1*, which was verified by immunoblotting (Fig. 5f), did not affect LDH release (Fig. 5g). However, cytosolic LPS-induced IL-1β release was not only dependent on caspase-4 and GSDMD (Fig. 5h), but also on the canonical NLRP3 inflammasome (Fig. 5i). oxPAPC also inhibited cytosolic LPS-induced cytotoxicity in primary human macrophages primed with Pam3CSK4 (Fig. 5j) or poly(I:C) (Fig. 5k), HMGB1 release and GSDMD cleavage (Fig. 5l), and this response was defective in CASP4 siRNA-silenced cells (Fig. 5m, n). Our results indicate that oxPAPC prevents caspase-4-mediated pyroptosis in human macrophages.

**oxPAPC competes with LPS for caspase binding**. Non-canonical inflammasome caspases directly bind to LPS, which causes their oligomerization and activation[13]. To test whether the inhibitory effect of oxPAPC is mediated by interfering with the caspase-LPS interaction, we first tested whether oxPAPC can bind to non-canonical caspases. Indeed, we were able to pull down transiently transfected Flag-tagged caspase-11 from cell lysates with immobilized oxPAPC, but the oxPAPC/caspase-11 interaction was lost in the presence of excess LPS (Fig. 6a), indicating that oxPAPC and LPS compete for caspase-11 binding. We next investigated oxPAPC binding to endogenous caspase-11 in poly(I:C)-primed WT, *Casp1*⁻/⁻*Casp11*⁻/⁻, and *Casp11*⁻/⁻ BMDMs. oxPAPC was able to precipitate caspase-11 from lysates of WT cells, but not from *Casp1*⁻/⁻*Casp11*⁻/⁻ and *Casp11*⁻/⁻ cells, or in the presence of excess LPS (Fig. 6b). We obtained comparable results in caspase-4-transfected HEK293 cell lysates (Fig. 6c), indicating a conserved mechanism. As both LPS and oxPAPC bound caspase-11, we tested whether oxPAPC could compete with LPS for caspase-11 binding. In agreement with previous results[13], biotinylated LPS purified Flag-tagged caspase-11 from transfected HEK293 cell lysates, but co-incubation with oxPAPC or unlabeled LPS prevented the LPS-caspase-11 interaction (Fig. 6d). Biotinylated LPS also co-purified caspase-11 from poly(I:C)-primed BMDM lysates and oxPAPC treatment competed with LPS binding to caspase-11 (Fig. 6e). This suggested a similar mechanism as the oxPAPC-mediated inhibition of TLR4

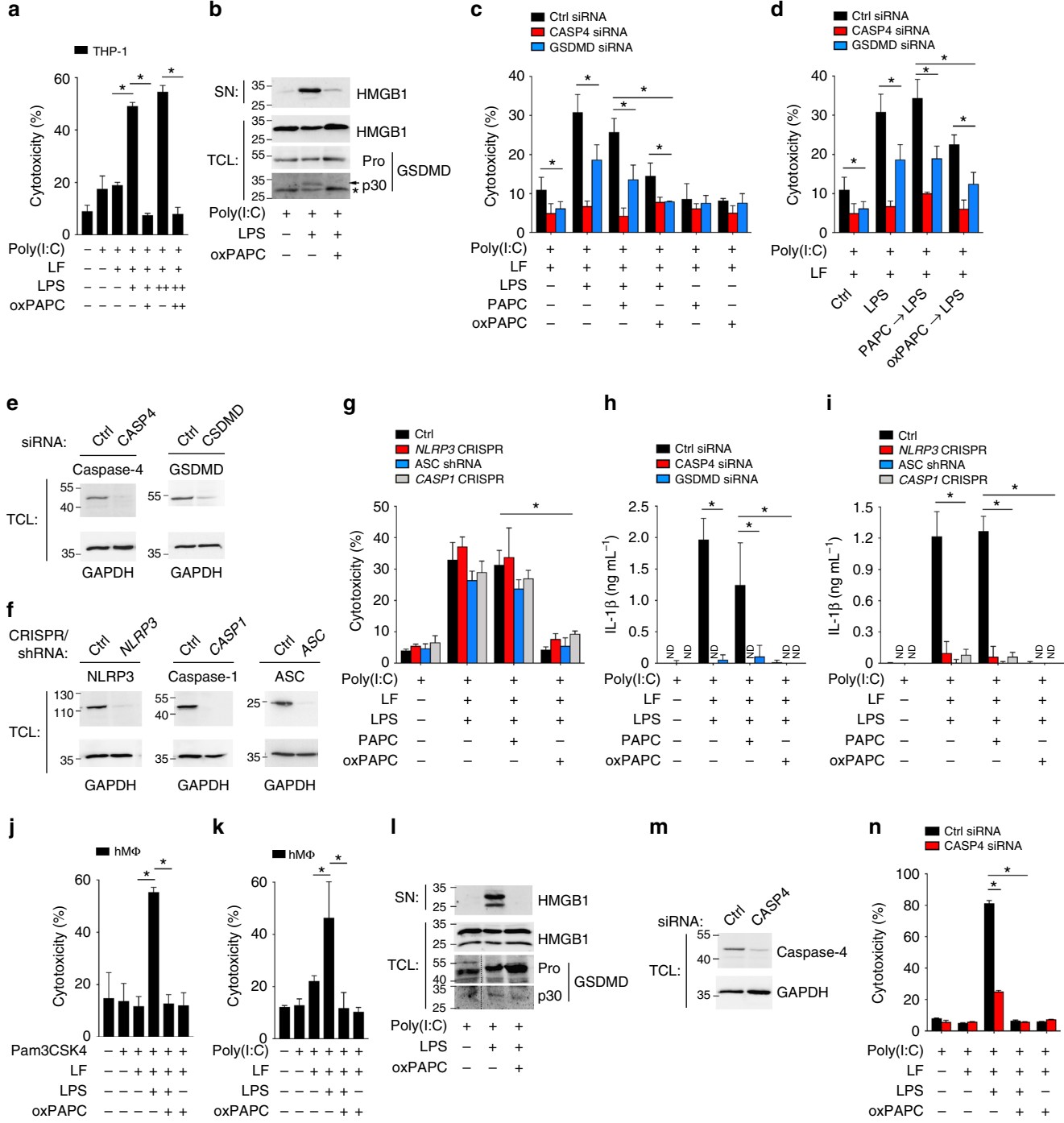

**Fig. 5** OxPAPC inhibits LPS-induced pyroptosis in human macrophages. **a, b** THP-1 cells were primed with poly(I:C) (1 µg ml⁻¹) for 6 h and then transfected with Lipofectamine 2000 (LF) and **a** LPS (+, 50 ng per well; ++, 100 ng per well) and oxPAPC (+, 100 ng per well; ++, 200 ng per well), or **b** LPS (50 ng per well) and oxPAPC (100 ng per well) as indicated. **a** LDH release in supernatants (SN) was determined after 4 h and presented as % cytotoxicity compared with LDH release in lysed cells. **b** SN and total cell lysates (TCL) were analyzed by immunoblot for HMGB1 and GSDMD, showing the pro-form (Pro) and cleaved (p30) GSDMD. *A cross-reactive protein. **c–e, h** THP-1 cells were transfected with control (Ctrl), CASP4, or GSDMD-specific siRNAs. **c, h** Cells were then primed and transfected as indicated or **d** pre-treated with either PAPC or oxPAPC for 30 min, followed by transfection of LPS and **c, d** culture SN analyzed for LDH release as above or **h** IL-1β release by ELISA. Cells were analyzed for **e** caspase-4 and GSDMD expression by immunoblot. **f, g, i** THP-1 cells with shRNA-silenced ASC or CRISPR-mediated knockout of NLRP3 and CASP1 were **f** analyzed for NLRP3, ASC, and caspase-1 expression by immunoblotting. Cells were transfected as above and culture SN analyzed for **g** LDH release and **i** IL-1β release. **j–n** Human primary macrophages (hMΦ) were primed with poly(I:C) (1 µg mL⁻¹) or Pam3CSK4 (1 µg mL⁻¹) for 6 h and then transfected with LPS and oxPAPC as above. **j, k** LDH release and **l** HMGB1 and GSDMD processing by immunoblot were determined as above. **m, n** Human primary macrophages were transfected with control (Ctrl) or CASP4-specific siRNAs, **m** caspase-4 expression determined by immunoblotting, and **n** cells were then primed and transfected as indicated and analyzed for LDH release as above. Data are representative of at least three independent experiments of at least triplicate samples. Error bars indicate ± SD; *P < 0.05 by two-tailed unpaired t-test

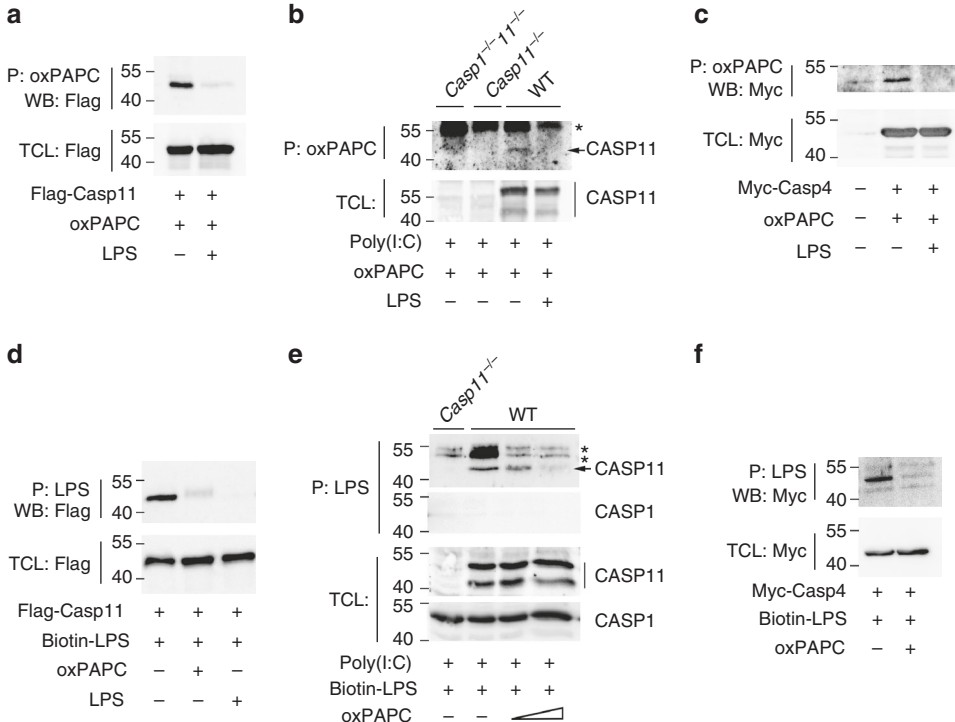

**Fig. 6** OxPAPC binds and competes with LPS for binding non-canonical caspases in macrophages. **a–c** (**a**, **c**) HEK293 cells transfected with a **a** Flag-tagged caspase-11 or **c** Myc-tagged caspase-4, or **b** 16 h poly(I:C) (1 μg mL$^{-1}$) primed BMDMs were lysed and total cell lysates (TCL) were incubated with oxPAPC (10 μg) or oxPAPC (10 μg) and LPS (100 μg) for 2 h, and purified (P) with an immobilized anti-oxPAPC antibody. Immune complexes and TCL were analyzed by immunoblot (WB) as indicated. **d–f** (**d**, **f**) HEK293 cells were transfected with a **d** Flag-tagged caspase-11 or **f** Myc-tagged caspase-4 or **e** BMDMs were primed as in **b**, lysed, and TCL were incubated with biotinylated LPS (1 μg) in the absence or presence of **d** oxPAPC (100 μg) or **e** oxPAPC (25–100 μg), or **f** either oxPAPC (100 μg) or unlabeled LPS (100 μg) for 2 h and purified with immobilized NeutrAvidin. Immune complexes and TCL were analyzed by immunoblot as indicated. *A cross-reactive protein. Data are representative of at least three independent experiments. A molecular weight marker (kDa) is indicated

signaling, where oxPAPC binds to the LPS sensing TLR4 co-receptors CD14 and LPS-binding protein[25]. Similar to mouse caspase-11, human caspase-4 also directly binds LPS[13]. Biotin-conjugated LPS also co-purified caspase-4 from myc-caspase-4-transfected HEK293 cell lysates, which was completely prevented in the presence of oxPAPC (Fig. 6f). This indicates that the oxPAPC-mediated inhibition of LPS binding to non-canonical inflammasome caspases occurs in human and mice, and that oxPAPC directly binds to non-canonical caspases and competes with LPS binding to inhibit pyroptosis.

**LPS and oxPAPC binding requires CARD and caspase domains**. As oxPAPC and LPS compete for binding, we next determined the necessary domains. Utilizing Flag-tagged caspase-11, caspase-11$^{CARD}$, caspase-11$^{\Delta CARD}$, and caspase-11$^{C254A}$ catalytic domain mutant allowed us to co-purify caspase-11 and caspase-11$^{C254A}$ with immobilized oxPAPC, but not caspase-11$^{CARD}$ and caspase-11$^{\Delta CARD}$ (Fig. 7a), suggesting that the caspase and CARD domains, but not the catalytic activity, were required for oxPAPC binding. An earlier study showed oxPAPC binding to caspase-11$^{\Delta 1-59}$, consisting of the C-terminal 30 amino acids of the CARD and the caspase domain[26]. As oxPAPC competed with LPS binding, we hypothesized that both molecules may bind to the same domain. However, an earlier study demonstrated that LPS binds to the CARD of caspase-11[13]. Using our mutants, we observed an identical binding pattern for caspase-11 to LPS as for caspase-11 to oxPAPC, as immobilized LPS also co-purified caspase-11 and caspase-11$^{C254A}$ from cell lysates, but failed to co-purify caspase-11$^{CARD}$ and caspase-

11$^{\Delta CARD}$ (Fig. 7b), suggesting that both domains were also necessary for LPS binding. To further specify the requirements for oxPAPC and LPS binding to caspase-11, we mutated positively charged residues within the CARD (mutants CARD-M1, M2, and M3) or in the caspase domain (mutants CD-M1, M2, and M3), due to the negative charge of LPS and oxPAPC, and used the catalytic domain C254A mutant as a template to prevent autocatalytic cleavage. All CARD mutants failed to purify LPS, although CARD-M2, with mutated positively charged amino acids 38–55, retained very weak binding, indicating that several motifs within the CARD contribute to LPS binding. In addition, although positively charged residues from amino acids 103 to 185 (CD-M1) and from 310 to 360 (CD-M3) were dispensable for LPS binding, positively charged residues from 220 to 294 (CD-M2) also contributed to LPS binding (Fig. 7c). This corroborated our domain deletion studies and revealed that, contrary to earlier studies, positively charged amino acid residues within the CARD and the caspase domain were required for efficient LPS binding, and that neither domain alone was sufficient for binding. Some of the caspase domain mutants showed increased molecular weight by SDS-polyacrylamide gel electrophoresis, which did not correspond to the amino acid sequence, suggesting potential modification, which did not impair LPS binding. Next, we utilized the same mutants to map oxPAPC binding. Interestingly, only K19E (CARD-M1) and positively charged residues throughout the caspase domain were required for oxPAPC binding, as all caspase domain mutants failed to purify oxPAPC (Fig. 7d). Hence, even though positively charged residues in the CARD and the caspase domain are crucial for LPS and oxPAPC binding to Caspase-11, there was distinct preference for the positively charged residues

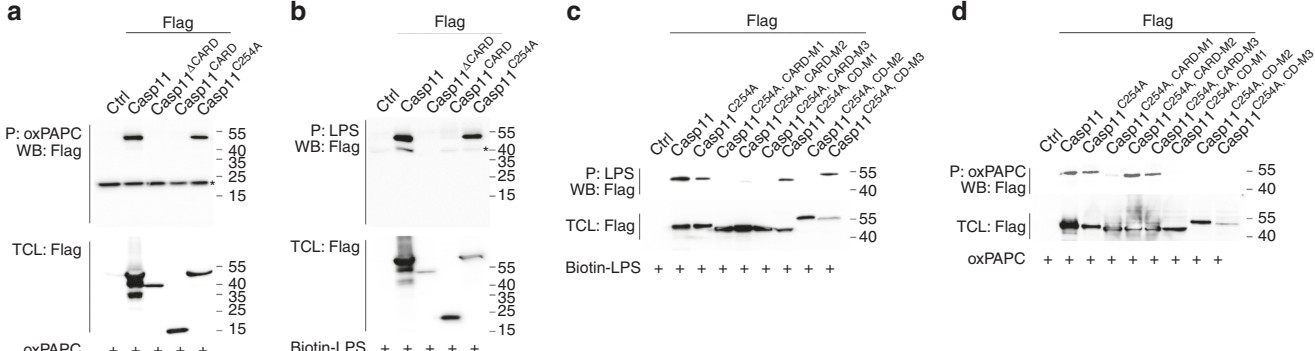

**Fig. 7** LPS and oxPAPC binding requires both the CARD and caspase domain. **a**, **b** HEK293 cells were transfected with Flag-tagged empty vector, caspase-11, caspase-11$^{\Delta CARD}$, caspase-11$^{CARD}$, and caspase-11$^{C254A}$, lysed, and total cell lysates (TCLs) were incubated with **a** oxPAPC (10 μg) for 2 h and purified (P) with an immobilized anti-oxPAPC antibody or **b** biotinylated LPS (1 μg) for 2 h and purified with immobilized NeutrAvidin. Immune complexes and TCL were analyzed by immunoblot (WB) as indicated. **c**, **d** HEK293 cells were transfected with the indicated caspase-11 mutants, lysed, and TCL were incubated with **c** biotinylated LPS (1 μg) or **d** oxPAPC (10 μg), and analyzed as above. Data are representative of at least three independent experiments. A molecular weight marker (kDa) is indicated

that are involved in LPS and oxPAPC binding to caspase-11, which we predict is responsible for the different activities of LPS and oxPAPC.

**oxPAPC ameliorates acute LPS-induced shock in vivo.** *Tlr4*$^{-/-}$ and *Casp11*$^{-/-}$ mice are protected to a varying degree from LPS-induced lethal septic shock[2,9,39,49]. However, upon priming with a TLR3 instead of a TLR4 ligand, *Casp11*$^{-/-}$ mice, but not *Tlr4*$^{-/-}$ mice, are partially protected from LPS-induced lethality[8,9,50,51]. A recently established prime and challenge sepsis model demonstrated a significant contribution of caspase-11 in the septic response with a 50–65% survival rate in *Casp11*$^{-/-}$ mice[8,9,50,51]. This rapid LPS-mediated shock is in contrast to LPS challenge-induced sepsis, which requires 20 times more LPS and takes 24–48 h to cause mortality in mice. We utilized the prime and challenge sepsis model developed by Miao and colleagues[9,51] to allow very low LPS doses to cause largely caspase-11-dependent acute LPS-induced shock. To validate that this response is dependent on caspase-11 and not the canonical NLRP3 inflammasome, we primed WT, *Nlrp3*$^{-/-}$, and *Casp11*$^{-/-}$ mice by intraperitoneal (i.p.) injection with poly(I:C) for 7 h and then i.p. injected 5 μg per kg body weight of ultrapure LPS, which resulted in rapid mortality of WT and *Nlrp3*$^{-/-}$ mice within 4 h, whereas 70% of *Casp11*$^{-/-}$ mice were protected (Fig. 8a). To determine the effect of oxPAPC specifically in this partially caspase-11-dependent septic shock model, we primed WT mice with poly(I:C) as above, then i.p. injected either phosphate-buffered saline (PBS), PAPC, or oxPAPC, followed by i.p. injection of 5 μg per kg body weight of ultrapure LPS 10 min later. Although PBS/LPS- and PAPC/LPS-injected mice rapidly deceased, 50% oxPAPC/LPS-injected mice were protected, which is close to that of *Casp11*$^{-/-}$ mice (Fig. 8b). Owing to the very rapid mortality, we reduced the injected LPS to 2.5 μg per kg body weight. PBS/LPS-injected mice displayed rapid hypothermia starting within 45 min of LPS injection, averaging 27 °C after only 105 min, whereas the body temperature of oxPAPC/LPS-injected mice did not drop below 34 °C (Fig. 8c). This ameliorated septic response in oxPAPC/LPS-injected mice was further emphasized by the significant reduction of serum tumor necrosis factor (TNF) levels (Fig. 8d), which is a key marker of sepsis. HMGB1 is released during sepsis and is sufficient to recapitulate LPS-induced lethality upon injection[52]. HMGB1 and cleaved GSDMD p30 release into the peritoneal cavity were significantly reduced in oxPAPC/LPS-injected mice compared with PBS/LPS-injected mice (Fig., 8e, f). These results clearly indicated that oxPAPC ameliorated

caspase-11-mediated pyroptosis and the resulting mortality in vivo. Although we were able to reproduce the previously reported partial contribution of caspase-11 in this model, we could not validate a TLR4-independent response[9], as our colony of *Tlr4*$^{-/-}$ mice were protected from sepsis in this model (Supplementary Fig. 12a). Recent evidence revealed that LPS enhances the activity of insulin present in Opti-MEM resulting in a lethal hypoglycemic shock that depends on caspase-11, TLR4, and complement, rather than just caspase-11[51]. Although no completely caspase-11-dependent in vivo response has been identified, lethality in response to high-dose LPS after TLR2 or TLR3 priming is caspase-11, but not TLR4 dependent[8,15]. We confirmed that LPS-induced lethality after poly(I:C) priming is TLR4-independent at LPS concentrations above, but not below 25 mg kg$^{-1}$ (Supplementary Fig. 12b). We primed mice with poly(I:C) as above, followed by i.p. injection of 25 mg kg$^{-1}$ LPS, which caused lethality in WT mice within 24 h. However, 20% of *Casp11*$^{-/-}$ mice survived. Significantly, pre-injection of oxPAPC, but not PAPC, protected WT mice comparable to *Casp11* deficiency (Fig. 8g). Although survival of *Tlr4*$^{-/-}$ mice was prolonged for up to 48 h, pre-injection of oxPAPC, but not PAPC, also protected *Tlr4*$^{-/-}$ mice to an even higher degree than WT mice (Fig. 8h), likely to be due to the combined protective effect of *Tlr4* deficiency and oxPAPC-mediated inhibition of caspase-11. *Tlr4*$^{-/-}$ mice pre-injected with oxPAPC, but not PAPC or PBS, also displayed reduced LPS-induced hypothermia (Fig. 8i), demonstrating that oxPAPC ameliorates acute LPS-induced shock in a TLR4-independent manner by inhibiting caspase-11.

**Discussion**

Caspase-11 contributes to LPS-induced lethality and, similarly, caspase-4 senses LPS and can functionally replace caspase-11 during septic shock[14,47,49]. To maintain homeostasis, checkpoints must be in place to prevent inappropriate non-canonical inflammasome activation. However, unlike the complex upstream regulatory complex present in canonical inflammasomes and the presence of endogenous inhibitory proteins[53], non-canonical inflammasome caspases directly function as receptor and effector, and no regulatory mechanisms have been described. Although a Toll-like receptor (TLR)-mediated transcriptional checkpoint for caspase-11 expression is necessary, human caspase-4 is constitutively expressed[31,54]. Here we introduce oxPAPC as a novel TLR4-independent checkpoint regulator for non-canonical inflammasomes in human and mouse macrophages. oxPAPC is present in circulation and is increased during

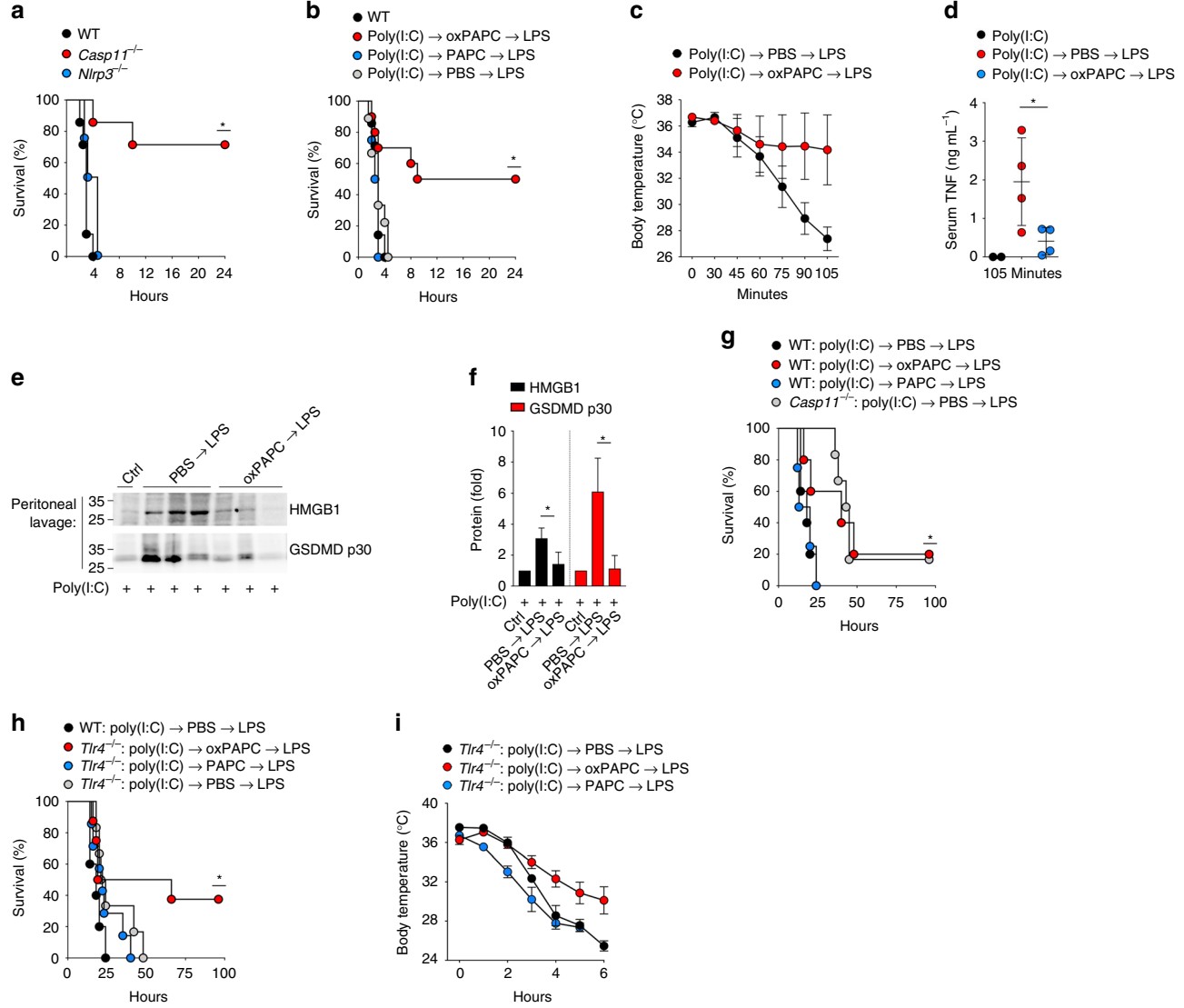

**Fig. 8** oxPAPC ameliorates acute LPS-induced shock in vivo. **a** WT, Casp11[−/−], and Nlrp3[−/−] mice (n = 7) were i.p. injected with poly(I:C) (10 mg kg[−1]) and 7 h later i.p. injected with LPS (5 μg kg[−1]) in Opti-MEM and survival was determined. **b** WT mice (n = 10) were i.p. injected with poly(I:C) (10 mg kg[−1]) and 7 h later i.p. injected with either PBS, oxPAPC (1 mg kg[−1]), or PAPC (1 mg k[−1]), and 10 min later i.p. injected with LPS (5 μg kg[−1]) and survival was determined. **c–f** WT mice (n = 4) were primed with poly(I:C) as above, 7 h later i.p. injected with either PBS or oxPAPC (500 μg kg[−1]), and 10 min later i.p. injected with LPS (2.5 μg kg[−1]) in Opti-MEM. **c** Body temperature was determined at the indicated times by rectal probe, **d** serum TNF was determined by ELISA, and **e** peritoneal HMGB1 and p30 GSDMD by immunoblot, and **f** HMGB1 and GSDMD protein quantified and presented as fold compared with PBS-injected mice. **g–i** WT, Tlr4[−/−], or Casp11[−/−] mice (n = 5-9) were primed with poly(I:C) as above and 7 h later i.p. injected with either PBS, PAPC (250 mg kg[−1]), oxPAPC (250 mg kg[−1]), and 10 min later i.p. injected with LPS (25 mg kg[−1]) in PBS, and **g, h** survival and **i** body temperature was determined as above. Data were analyzed by **a, b, g, h** asymmetrical Log-rank Mantel–Cox survival test, *P < 0.05; **c, i** two-way ANOVA + Bonferoni post test. Error bars indicate ± SD; *P < 0.05; **d, f** two-tailed unpaired t-test and error bars indicate ± SD; *P < 0.05. A molecular weight marker (kDa) is indicated

inflammation and infection[20,23,32,55]. Therefore, PAPC peroxidation products may raise the threshold required for non-canonical inflammasome activation to prevent minor amounts of LPS from triggering a cytotoxic response. oxPAPC has been shown earlier to prevent TLR4-mediated transcriptional responses[21–23,25], but some evidence also supports a proinflammatory role of oxPAPC by activating TLR4[19,20]. Recently, oxPAPC has been described to promote a weak agonistic activity in DCs, but not macrophages, by promoting a caspase-11, caspase-1, ASC, and NLRP3-dependent release of IL-1β to potentiate LPS-mediated T-cell activation[26]. However, this response contradicts several in vivo studies showing that oxPAPC prevents rather than augments cytokine release[23,25], and that oxPAPC actually

prevents DC activation, maturation, cytokine release, and T-cell stimulation[29].

Our studies in macrophages compellingly demonstrate an inhibitory role of oxPAPC on cytosolic LPS responses, and that this mechanism is conserved between mice and humans. oxPAPC not only blocked pyroptosis, but also IL-1β release downstream of caspase-11, mediated by the canonical NLRP3 inflammasome. Despite TLR3 priming, IL-1β release was partially TLR4 dependent, indicating that cytosolic delivery of LPS in liposomes engaged cell surface TLR4 to promote Il1b transcription. However, contrary to TLR4-dependent IL-6 release in response to extracellular LPS, cytosolic LPS-mediated IL-6 release was only partially dependent on TLR4, was not affected by oxPAPC, and was independent of the non-canonical and canonical

inflammasome. Importantly, oxPAPC-mediated inhibition of pyroptosis was limited to the non-canonical inflammasome, as NLRC4-mediated activation of caspase-1 was not inhibited by oxPAPC. We find a dose-dependent antagonistic function of oxPAPC in macrophages, but no agonistic activity at any tested concentration in primed or unprimed cells. Hence, oxPAPC functions differently in macrophages and DCs[26,29]. It is possible that individual phospholipids generated by PAPC peroxidation promote pro- or anti-inflammatory responses, and that macrophages and DCs display cell-type-specific sensitivities to these individual components.

Indeed, analysis of DCs revealed a cell-type-specific response to oxPAPC. DCs released some IL-1β in response to LPS priming alone and compared with nigericin, oxPAPC treatment even at high concentrations, showed only a weak agonistic activity and modest IL-1β release after prolonged 18 h treatment. In macrophages, oxPAPC inhibited LPS-induced caspase-11 activation at any tested concentration ranging from 0.2 to 10 µg mL$^{-1}$. However, neither LPS priming nor high oxPAPC concentrations up to 200 µg mL$^{-1}$ promoted IL-1β release. We already achieved potent caspase-11 inhibition with 0.6 µg mL$^{-1}$ (0.72 µM) and achieved complete inhibition with 2 µg mL$^{-1}$ (2.4 µM). In contrast, the weak response in DCs required at least 170-fold more oxPAPC (100–200 µg mL$^{-1}$; 120–240 µM). Oxidized phospholipids have been previously linked to NLRP3 activation, but also only when in excess of 250-fold higher concentrations were used[28]. The relevance of these high oxPAPC concentration-induced inflammatory responses will require additional studies, as oxPAPC concentrations of only 0.1–1 µM have been measured in mouse and human plasma, which may reach 10 µM locally[23,32,55] and are well within the range where we observe maximum caspase-11 inhibition, but not DC activation. Previous reports also revealed that much lower oxPAPC concentrations are required for the antagonistic function of oxPAPC on TLR4 signaling, than for induction of proinflammatory responses[20,23]. Of note is that another study reported that oxPAPC actually inhibits DC activation, co-stimulatory molecule expression, and cytokine release, and in this study, low oxPAPC concentrations of 5–30 µg mL$^{-1}$ (6–36 µM) were shown to be inhibitory[29]. Hence, at much higher concentrations and long incubation times, oxPAPC may eventually switch from promoting anti-inflammatory signaling to eliciting proinflammatory responses, which we, however, did not observe for macrophages in our study[23]. Such a proinflammatory response would also not be reconcilable with the sepsis protection elicited by oxPAPC in vivo.

However, we did not observe an inhibitory activity of oxPAPC in DCs, unless oxPAPC is artificially delivered to the cytosol by transfection. The reason for this different response is not known and both cell types have a non-canonical inflammasome and respond to LPS with pyroptosis and cytokine release. We ruled out potential oxPAPC batch differences as being responsible for the disparity in our studies, but we observed that DCs are much less efficient in intracellular oxPAPC uptake, when compared with macrophages. This qualitative difference may explain the primarily TLR4-mediated cell surface response observed in DCs, which then engages the canonical NLRP3 inflammasome, as reported earlier[26,28]. This may occur through TLR-dependent rapid inflammasome activation[56]. Forced delivery of oxPAPC into the cytosol of DCs results in a potent inhibition of caspase-11-dependent IL-1β secretion, rather than promoting it, which revealed that the inability of DCs to efficiently take up oxPAPC is a main reason for this different response of oxPAPC compared with macrophages. In contrast, macrophages efficiently take up oxPAPC, which may therefore largely bypass TLR4 and enable targeting caspase-11 directly in the cytosol. The reason for this quality difference in cellular uptake is not known, but we ruled

out CD36 and TLR4 as a receptor necessary for oxPAPC uptake in macrophages, and additional studies are necessary to determine the precise mechanism of oxPAPC uptake, as CD14 only partially affects oxPAPC uptake[27]. We further show that oxPAPC treatment and transfection yields a comparable inhibition in macrophages, but we used transfection to partially bypass TLR4-mediated responses to specifically engage caspase-11.

Although an initial study observed oxPAPC-mediated IL-1β release only in DCs, but not in macrophages[26], more recently the same group reported that very high concentrations of the purified oxPAPC lipids PGPC and POVPC are able to promote IL-1β release also in primed macrophages[27], similar to an earlier POVPC study[28]. We did not observe this response with these principal phospholipids generated by PAPC peroxidation, which showed a comparable non-canonical inflammasome inhibitory activity in macrophages, whereas DMPC, which was inactive in DCs[26], did not show any activity in macrophages as well. Hence, the same PAPC peroxidation products that elicit proinflammatory responses in DCs, inhibit caspase-11 in macrophages in our study, indicating a cell-type-specific response, but the differences in macrophages between our studies are not known. However, POVPC or PGPC represent < 10% of oxPAPC and, hence, the required concentrations to promote inflammatory responses described in these studies are not found in serum and it is possible that only non-physiologically high concentrations of oxPAPC components elicit proinflammatory responses, similar to what has been reported earlier for TLR4 responses[23].

Besides the question of the type of inflammatory response elicited by oxPAPC, the other obvious difference in our studies is the role of caspase-11 itself in the DC response to high concentrations of oxPAPC. Although Zanoni et al.[26] reported that this response is dependent on caspase-11 and the canonical inflammasome, in our side by side analysis, caspase-11 was completely dispensable for the weak LPS/oxPAPC-induced IL-1β release in FLT3L/GM-CSF and GM-CSF-differentiated BMDCs, which was absolutely dependent on TLR4 and the canonical NLRP3 inflammasome. Also contradicting a caspase-11 involvement, Zanoni et al.[26] reported that the catalytic activity of caspase-11 is dispensable for oxPAPC-mediated IL-1β release[26], although caspase-11-mediated activation of the canonical NLRP3 inflammasome for IL-1β release requires the catalytic activity of caspase-11 for Pannexin-1 cleavage, K$^+$ efflux, and NLRP3 activation[50]. Notably, the same group recently reported a partial caspase-11 dependence, but confirmed the complete dependence on the canonical inflammasome that we also observed[27]. oxPAPC-mediated activation of DCs is also at odds with the finding that oxPAPC prevents DC activation, maturation, and cytokine release[29], but may reflect the reported canonical NLRP3 inflammasome activation in response to very high concentrations of POVPC[28].

Our in vivo studies utilizing two LPS-induced shock models further supported an antagonistic rather than agonistic activity of oxPAPC during sepsis. We initially tested priming with a sub-lethal dose of poly(I:C) followed by induction of endotoxic shock with a low-dose LPS using Opti-MEM as an enhancer[9,51], as in this model low concentrations of oxPAPC should outcompete LPS. Seventy percent of Casp11$^{-/-}$ mice, but no WT or Nlrp3$^{-/-}$ mice, survived; however, oxPAPC injection in WT mice provided a 50% protection, demonstrating that oxPAPC protects from LPS-induced death that is partially mediated by caspase-11. However, contrary to the earlier report[9], Tlr4$^{-/-}$ mice derived in our animal facility were completely protected, even at LPS doses, where Casp11$^{-/-}$ mice did not show any protection, suggesting that both TLR4 and caspase-11 contribute to lethality in this model and these responses cannot be separated in vivo. This finding has just been validated by others as well[51]. We therefore

employed another caspase-11-dependent sepsis model using poly (I:C) priming, but a high dose of LPS[8]. In the absence of Opti-MEM, $Tlr4^{-/-}$ mice were only protected from LPS-induced death at doses below 25 mg per kg body weight, but 28% $Casp11^{-/-}$ mice still survived. Reported protection of $Casp11^{-/-}$ mice from sepsis varies and ranges from 20%[9] to 60%[8,50]. Importantly, oxPAPC pre-injection also protected 20% of $Tlr4^{-/-}$ mice, demonstrating a TLR4-independent role of oxPAPC in ameliorating lethal septic shock in mice.

Mechanistically, we provided evidence that oxPAPC competed with LPS binding to caspase-4 and caspase-11, demonstrating a conserved function and mechanism. However, although we could reproduce that the caspase-11 CARD is required for LPS binding, we observed that it is not sufficient, as it also required positively charged amino acid residues within the caspase domain. Importantly, binding of oxPAPC also required both the CARD and the caspase domain of caspase-11. As the catalytic domain was not required for oxPAPC binding, oxPAPC likely did not simply block or mask the catalytic domain, but could sterically hinder substrate interactions. Mutational analysis of positively charged amino acid residues further supported our finding that residues in both domains were necessary for oxPAPC binding. Interestingly, LPS binding also required overlapping, but also unique amino acid residues within both domains, which we predict are responsible for the functional differences of LPS and oxPAPC on caspase-11 activity. This only partially overlapping binding pattern further supported a competitive binding mechanism, which we also experimentally validated. Observed differences in binding may have resulted from using partially truncated mutants, omitting caspase domain mutants or recombinant vs. cell-based approaches to determine binding.

In summary, although oxPAPC is known to modulate inflammatory responses, its precise role has still been controversial[19–23,25,26,29]. Our findings support the known role of oxPAPC as an anti-inflammatory regulator and provide the first evidence for a negative regulation of the non-canonical inflammasome caspases by oxPAPC. Importantly, we demonstrated that this response is predominant in macrophages and is conserved between humans and mice. oxPAPC and its derivatives could therefore provide a basis for novel therapies targeting non-canonical inflammasomes during Gram-negative bacterial sepsis.

## Methods

**Mice**. C57BL/6 J WT, $Casp1^{-/-}Casp11^{-/-}$, $Casp11^{-/-}$, $Tlr4^{-/-}$, $Tlr3^{-/-}$, and $Cd36^{-/-}$ mice were obtained from the Jackson Laboratories, and $Casp1^{-/-}$, $Nlrp3^{-/-}$, and $Asc^{-/-}$ mice from Vishva M. Dixit (Genentech) and described earlier[15,43,57]. All mice were generated or were backcrossed to C57BL/6 for at least 10 generations and housed in a specific pathogen-free animal facility, and were derived from in-house breeding. All experiments were performed on age and gender-matched, randomly assigned 8–14-week-old mice conducted according to procedures approved by the Northwestern University Committee on Use and Care of Animals. The investigators were not blinded to the genotype of mice.

**Plasmids, reagents, and antibodies**. A caspase-4 complementary DNA was amplified by standard PCR from a human cDNA library and cloned into pcDNA3-Myc. pCMV-Flag-caspase-11 was obtained from Addgene (21145)[58]. Caspase-11$^{CARD}$ (aa 1–113), caspase-11$^{\Delta CARD}$ (aa 91–373), caspase-11$^{C254A}$, and the following on the catalytic domain C254A mutant: caspase-11$^{CARD-M1(K19E)}$, caspase-11$^{CARD-M2(K38, K42, K44, K53, R54E, W55A)}$, caspase-11$^{CARD-M3(K62, 63, 64E)}$, caspase-11$^{Caspase domain (CD)-M1(K103, R112, R115, K117, K125, R130, R132, K133, K143, R148, K159, K172, K185E)}$, caspase-11$^{CD-M2(K220, R243, K245M K247, R255, R265, K269, R274, R280, K288, K294E)}$, caspase-11$^{CD-M3(R310, K312, R321, R327, K328, K341, K348, R360, R365E)}$ were synthesized as gBlocks (IDT) and cloned into pcDNA3-Flag. All expression constructs were sequence verified. Ultra-pure unlabeled and biotinylated LPS (E. coli serotype O111:B4), Pam3CSK4, and low-molecular-weight (LMW) poly(I:C) were purchased from Invivogen. E. coli LPS serotype O55:B5 was purchased from Enzo Life Sciences. oxPAPC was purchased from Invivogen, Hycult Biotech, and Avanti Polar Lipids. oxPAPC used in most experiments were from Invivogen, unless indicated otherwise. PGPC, POVPC, and DMPC were from Cayman Chemical, and PAPC and 1-palmitoyl-2-arachidonoyl-*sn*-glycero-3-

phosphoethanolamine (PAPE) were from Avanti Polar Lipids. H. pylori GU2 LPS was from Wako. NeutrAvidin agarose was from Pierce (Thermo Scientific). Antibodies for pro-IL-1β (AF-401-NA, R&D Systems), mature IL-1β (52718, Cell Signaling), caspase-11 (17D9, Sigma), caspase-4 (4B9, Santa Cruz Biotechnology), caspase-1 (M-20, Santa Cruz), GSDMD (G7422, Sigma; H-11, Santa Cruz Biotechnology), HMGB1 (ab18256, Abcam), c-myc epitope (Santa Cruz Biotechnology), Flag M2 (Sigma) and biotinylated anti-oxidized phospholipid antibody (clone E06, Avanti Polar Lipids). For flow cytometry, the following fluorescent reagents were used: APC anti-mouse CD11c (N418, Biolegend), PE-Cy7 anti-mouse F4/80 (BM8, Biolegend), Fixable Viability Dye eFluor 506 (eBiosciences), and Alexa Fluor 488 and 647 streptavidin (Invitrogen). Full-length uncropped blots are presented in Supplementary Figure 13–15.

**Acute LPS-induced shock**. Mice were primed with poly(I:C) (LMW, 10 mg kg$^{-1}$, i. p.) for 7 h, followed by low-dose LPS challenge (2.5 or 5 µg kg$^{-1}$, i.p.) in 0.5 ml Opti-MEM (Invitrogen)[9]. Ten minutes before LPS injection, mice were i.p. injected with either PAPC or oxPAPC (500 µg kg$^{-1}$ or 1 mg kg$^{-1}$) or PBS. Alternatively, poly (I:C)-primed mice were i.p. injected with LPS (25 mg kg$^{-1}$, i.p.) in PBS and either PAPC or oxPAPC (250 µg kg$^{-1}$) as above. Rectal temperature was measured with a Micro Therma 2T thermometer (ThermoWorks). Blood was collected for determining serum TNF levels by ELISA (BD Biosciences). HMGB1 and GSDMD in peritoneal lavage fluids were detected by immunoblotting after trichloroacetic acid precipitation.

**Cell culture**. BM cells were flushed from femurs and tibias, and erythrocytes were lysed by sterile ACK lysing buffer (150 mM NH$_4$Cl, 10 mM KHCO$_3$, and 0.1 mM Na$_2$EDTA) at room temperature for 5 min. BMDCs were differentiated with either GM-CSF alone (20 ng mL$^{-1}$, R&D Systems) or the combination of FLT3L (100 ng mL$^{-1}$, Shenandoah Biotechnology) and GM-CSF (300 pg mL$^{-1}$) in RPMI 1640 medium supplemented with 10% heat-inactivated FBS (Invitrogen) and 50 µM 2-mercaptoethanol, following an adapted protocol from previous publications[59,60]. Two thirds of media supplemented with growth factor(s) was replaced on day 3, 5, and 7, and non-adherent cells were used on day 9 at a concentration of $1 \times 10^6$ cells per mL. DC purity was determined by flow cytometry as CD11c$^+$ cells. BMDMs were differentiated with M-CSF from CMG 14-12 cell conditioned medium[61] (20%) in Dulbecco's modified Eagle's medium (DMEM) medium, supplemented with 10% heat-inactivated FBS (Invitrogen), and 5% horse serum (Invitrogen), and analyzed after 7 days. $Gsdmd^{-/-}$ femurs were obtained from Dr T.-D. Kanneganti (St. Jude Children's Research Hospital). PMs were obtained by peritoneal lavage 3 days after i.p. injection of 1 mL 4% aged thioglycollate medium. Peripheral blood-derived human macrophages (hMΦ) were isolated from healthy donor blood after obtaining informed consent under a protocol approved by Northwestern University Institutional Review Board by Ficoll-Hypaque centrifugation (Sigma) and countercurrent centrifugal elutriation in the presence of 10 µg mL$^{-1}$ polymyxin B using a JE-6B rotor (Beckman Coulter), as described[62–65]. HEK293 and THP-1 cells were obtained from ATCC, and were routinely tested for mycoplasma contamination by PCR and cultured in complete DMEM and RPMI 1640 medium supplemented with 10% heat-inactivated FBS (Invitrogen).

**Transfection and treatment of macrophages and DCs**. Macrophages were plated in 96-well plates at $5 \times 10^4$ cells per well and primed with Pam3CSK4 (1 µg mL$^{-1}$, Invivogen) or poly(I:C) (LMW) (1 µg mL$^{-1}$, Invivogen) for 6 h, followed by transfection of ultra-pure E. coli LPS (0111:B4; Invivogen) (50 ng per well). Where indicated, H. pylori LPS (25–500 ng per well) or oxPAPC (10–1875 ng per well) was transfected with or without LPS. Transfection was performed in Opti-MEM (Invitrogen) using DOTAP (Biontex Labs, 750 ng) (for BMDMs and PMs) or Lipofectamine 2000 (Invitrogen, 250 ng) (for THP-1 cells and hMΦ). Plates were then centrifuged at $200 \times g$ for 5 min and incubated at 37 °C for 2 or 4 h as indicated. Poly(I:C) ($1 \times 10^6$) or Pam3CSK4 primed cells were electroporated with 1 µg ultra-pure LPS (0111:B4; Invivogen) and 2 µg oxPAPC, using the Neon Transfection System (Invitrogen, setting: 1600 V, 10 ms, 3 pulses). Recombinant flagellin from S. typhimurium (Invivogen) (15 ng per well) was transfected with or without oxPAPC into Pam3CSK4-primed BMDMs using DOTAP and incubated at 37 °C for 4 h. BMDCs were primed with LPS (O55:B5, 1 µg mL$^{-1}$) for 3 h, followed by treatment with oxPAPC (5–200 µg mL$^{-1}$) and culture supernatants were analyzed 18 h after LPS treatment[26]. For LPS and oxPAPC transfection, 750 ng DOTAP suspended in 10 µL Opti-MEM and 0.5 or 5 µg LPS or 1 and 10 µg of oxPAPC suspended in 10 µL Opti-MEM were allowed to equilibrate at room temperature for 5 min, mixed, and incubated at room temperature for 30 min, and added to 100,000 LPS-primed cells in 100 µL. Plates were then centrifuged at $200 \times g$ for 5 min and culture supernatants analyzed after 15 h.

**Bacteria**. The following S. typhimurium SL1344-based mutants $\Delta sifA$, $\Delta sifA/flgB$ ($flgB$::tn10dTet)[5] and $\Delta$SPI-1: ($orgA$::Tet), $\Delta$SPI-1/$\Delta fla$: ($orgA$::Tet, $fljAB$::Kan, $fliC$::Cm), and $\Delta$SPI-1/$\Delta$SPI-2: ($orgA$::Tet, $ssaV$::Kan)[5,45,66] were grown to stationary phase overnight in Luria-Bertani (LB) broth at 37 °C with aeration. BMDMs ($4.5 \times 10^4$) were infected at a multiplicity of infection of 25:1 and centrifuged for 5 min at $200 \times g$, to ensure comparable adhesion of the bacteria to the cells. Thirty minutes post infection, cells were washed twice with PBS, followed by incubation in

Gentamycin (Amresco, 50 µg mL$^{-1}$)-containing media, and cells were incubated for 3 h.

**Outer membrane vesicles**. *E. coli* OMV were isolated as previously described[7]. Briefly, *E. coli* BL21 were grown in 500 mL LB until the cultures reached an OD$_{600}$ of 0.6 and the bacteria-free supernatant was collected by centrifugation at 10,000 × *g* for 15 min at 4 °C. This supernatant was further filtered through a 0.45 µm filter and then centrifuged at 38,000 × *g* for 2 h at 4 °C. The vesicle-containing pellet was resuspended in 1 mL fresh PBS, followed by ultracentrifugation at 100,000 × *g* for 1.5 h at 4 °C. The final pellet was resuspended in 100 µl PBS. The protein and LPS content of purified OMVs were quantified by Pierce BCA protein assay kit and LAL Chromogenic Endotoxin Quantitation Kit (Thermo Scientific), respectively.

**Cytotoxicity assay and cytokine measurement**. Culture supernatants were collected at the indicated time points. LDH in culture supernatants was measured using the LDH Cytotoxicity Detection Kit (Clontech). Cytotoxicity was defined as the percentage of released LDH compared with maximal LDH activity after cell lysis with 1% Triton X-100. IL-1β, IL-6, and TNF (OptEIA, BD Biosciences) and IL-1α (Ready-SET-Go, eBiosciene) were quantified in culture supernatants by ELISA.

**Quantitative real-time PCR**. Total RNA was isolated from cells using the E.Z.N.A. Total RNA isolation Kit (Omega Bio-tek) and reverse-transcribed (Verso cDNA Synthesis Kit, Thermo Scientific). Gene expression analysis was performed on an ABI 7300 Real-Time PCR Machine (Applied Biosystems) and displayed as relative expression compared to β-Actin, using FAM-labeled mouse primers for *Il1b* (Mm00434228_m1, Invitrogen) in combination with VIC-labelled primers for *Actb* (Mm00607939_s1, Invitrogen).

**Gene silencing**. Silencer select siRNAs targeting CASP4 (s2413, s2414, and s2415) were purchased from Ambion (Thermo Scientific) and siRNA targeting GSDMD (5′-gtgtgtcaacctgtctatcaa-3′) was synthesized by IDT. THP-1 cells were electro-porated (Neon transfection system; Invitrogen) and human primary macrophages were transfected with siRNAs by Lipofectamine 2000 (Invitrogen). Forty-eight hours post transfection, cells were primed and transfected as described above. shRNA ASC-silenced THP-1 cells were described earlier[67]. NLRP3 and CASP1 knockout THP-1 cells were generated by CRISPR/Cas9 using pSpCas9(BB)-2A-GFP (for *NLRP3*; 5′-gctaatgatcgacttcaatg-3′) and lentiCRISPR (for *CASP1*; 5′-ttatccgttccatgggtga-3′) plasmids and the packaging plasmids pMD.2 G and psPAX2 (Addgene plasmids 12259 and 12260)[68,69].

**LPS and oxPAPC-binding assay**. HEK293 cells were transfected with mammalian expression constructs for myc-tagged pro-caspase-4 or Flag-tagged pro-caspase11 in six-well plates using Lipofectamine 2000 (Invitrogen). Thirty-six hours post transfection, cells were lysed in 50 mM HEPES (pH 7.4), 150 mM NaCl, 2 mM EDTA, 10% glycerol, and 1% Triton X-100, supplemented with protease inhibitors (Roche) for 30 min. For LPS binding, total cell lysates (TCLs) were incubated with 1 µg ultra-pure biotinylated *E. coli* LPS (0111:B4; Invivogen) in the presence or absence of 100 µg oxPAPC (Invivogen) or unlabeled LPS for 2 h at room temperature and purified with immobilized NeutrAvidin (Thermo Scientific), washed three times in binding buffer, eluted by Laemmli buffer, and analyzed by immunoblotting using anti -c-myc (Santa Cruz) and anti-Flag antibodies (Sigma), and horseradish peroxidase-conjugated secondary antibodies, ECL detection (Thermo Scientific), and image acquisition (Ultralum). TCLs (5%) were also analyzed as indicated. For oxPAPC binding, TCL were incubated with 10 µg oxPAPC in the presence or absence of 100 µg LPS for 2 h and pulled down by 1 µg biotinylated E06 antibody, immobilized onto NeutrAvidin beads, then analyzed as described above. For endogenous caspase binding, BMDMs were primed with poly(I:C) for 16 h and TCLs were analyzed as described above for caspase-11 (Sigma).

**Oxidized PAPE-N-biotin synthesis**. PAPE (Avanti Polar Lipids) was biotinylated as previously described[70] under nitrogen for 12 h at room temperature. The biotinylated phospholipid was then resuspended in chloroform, aliquoted in 1 mg per 13 × 100 mm glass test tube, evaporated the solvent under nitrogen using MULTIVAP nitrogen evaporator (Organomation), and left exposed to air for oxidation for 72 h.

**LPS and oxPAPC cellular uptake by flow cytometry**. BMDMs were primed with poly(I:C) (LMW, 1 µg mL$^{-2}$) for 6 h, followed by transfection of biotinylated LPS (O111:B4, Invivogen) with or without oxPAPC (Invivogen) using the same protocol and concentrations described above. Thirty minutes post transfection, cells were washed twice with cold PBS, fixed, permeabilized, and stained with Alexa Fluor 488 streptavidin. oxPAPC uptake in BMDMs and BMDCs was accessed 30 min after treatment with oxidized PAPE-N-biotin with the indicated concentration. Cells were washed twice with cold PBS, fixed, permeabilized, and stained with Alexa Fluor 488 streptavidin. Cells (0.5 × 10$^6$) were washed twice with cold PBS. Cell viability was assessed by incubation in Fixable Viability Dye eFluor 506 (eBioscience) for 30 min in the dark at 4 °C, followed by a single wash in 1 × PBS.

For surface staining, cells were incubated in 0.5 µg Fc Block (BD Biosciences) for 15 min at 4 °C, followed by staining with the indicated antibodies in autoMACS Running Buffer (Miltenyi Biotec) in the dark for 30 min at 4 °C. Cells were then washed twice with cold autoMACS Running Buffer, followed by fixation in 2% paraformaldehyde (Electron Microscopy Sciences). For oxPAPC and LPS uptake assays, cells were fixed and permeabilized using Cytofix/Cytoperm (BD Biosciences) for 20 min at 4 °C, washed twice with 1 × Perm/Wash buffer (BD Biosciences), and then incubated with indicated fluorescent conjugated streptavidin in 1 × Perm/Wash buffer for 30 min at 4 °C, followed by two washes in Perm/Wash buffer and a single wash in autoMACS Running Buffer, and resuspended in autoMACS Running Buffer for analysis. Sorting and data collection were performed using an LSRII instrument (BD Biosciences) and data were analyzed with FlowJo software (TreeStar, Inc.).

**Statistical analysis**. Graphs represent one representative experiment of three or more repeats, each consisting of at least triplicate samples, and are presented as the mean ± SD. A standard two-tailed unpaired *t*-test was used for statistical analysis of two groups with all data points showing a normal distribution; symmetrical Log-rank Mantel–Cox analyses were used to investigate differences in survival; and differences in body temperature was analyzed by two-way analysis of variance + Bonferoni post test. Values of $P < 0.05$ were considered significant (Prism 7, GraphPad). Sample sizes were selected based on preliminary results to ensure a power of 80% with 95% confidence between populations.

**Data availability**. All data that support the findings of this study are available from the corresponding authors upon request.

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

## Acknowledgements

We thank Dr Bo Shi for support with the elutriation of human primary macrophages, Drs Carla M. Cuda and Qi-Quan Huang for advice in BMDC differentiation, Dr. Joan M. Cook-Mills for the MULTIVAP nitrogen evaporator, and members of the Stehlik lab for helpful discussions. *Casp1*−/−, *Asc*−/−, and *Nlrp3*−/− mice were kindly provided by Dr Vishva M. Dixit (Genentech, USA), *Gsdmd*−/− femurs by Dr Thirumala-Devi Kanneganti (St. Jude Children's Research Hospital, USA), mutant *S. typhimurium* strains by Drs Edward A. Miao (University of North Carolina at Chapel Hill, USA), Samuel I. Miller (University of Washington, USA), and Denise M. Monack (Stanford University, USA), Plasmids pMD2.G and psPAX2 by Didier Trono (École Polytechnique Fédérale de Lausanne, Switzerland), and CMG 14-12 cells by Sunao Takeshita (National Center for Geriatrics and Gerontology Japan). This work was supported by the National Institutes of Health (AI099009, AR064349, and AI120618 to C.S., AI120625 to C.S. and A.D., and AR066739 to A.D.), a Cancer Center Support Grant (CA060553), and the Skin Disease Research Center (AR057216) to C.S. L.H.C. was supported by the Vietnam Education Foundation Fellowship and the American Heart Association (AHA, 15PRE25700116), and M.I. was supported by the AHA (15POST25690052).

## Author contributions

L.H.C. designed and performed most of the experiments, analyzed the data, and performed statistical analyses. M.I. supported the in vivo experiments and flow cytometry. R.

A.R. contributed to the BMDC studies. A.G. generated and analyzed the *NLRP3* KO THP-1 cells. E.P.M. supported the caspase-11 mutagenesis studies. D.M.M. provided key reagents and advice. A.D. and C.S. conceived the project and provided overall direction. All authors discussed the results, contributed to the writing, and approved the manuscript.

## Additional information

**Competing interests:** The authors declare no competing interests.

