## [Peer Review File · Nature Communications]

Reviewers' comments:

Reviewer #1 (Remarks to the Author):

The authors present a much-improved version of their manuscript on OxPAPC-mediated inhibition of caspase-11 and caspase-4. They have addressed most of my previous concerns/question, except the issue of addressing the discrepancy between their results and the publication by the Kagan lab (Zanoni et al.) (see point 1). The authors also add a wealth of new data, which raise some new questions (see below) that should however be easily addressed before publication.

1) The authors argue that both manuscript focus on different aspects of immunity, i.e. Zanoni on adaptive immunity and DCs, and their manuscript on macrophages and sepsis. Although this is partially correct, it is obvious that the real focus of both manuscripts is the effect of OxPAPC on the non-canonical inflammasome. Here the messages are diametrically opposed, and I fear that this will be a major source of cause of confusion once the current manuscript is published. The reason for these conflicting results might lie in the concentrations used (high vs. low concentration), the cell types (DCs vs. macrophages) and the source of the OxPAPC. Although both studies use the same source of OXPAPC, and the authors now include yet OxPAPC from other manufacturers, I am worried if batch-to-batch differences account for the conflicting results. If this would be the case, it would be a serious problem for all studies with OxPAPC since such context dependent results are hard to repeat and even harder to interpret.

To address this point, I would therefore again ask the authors to test their OxPAPC in BMDMs, using concentrations in the range used by the Kagan paper and determine if in DCs OxPAPC indeed works as an activator of the caspase-11 pathway.

2) Fig. 1b: Please provide the p-values for the difference between P3C4/LPS/DOTAP and P3C4/LPS/DOTAP+PAPC. Is this really not a significant reduction?

3) Fig. 1d: In the text the authors claim that Wt and TLR3 KO show 'comparable' level of cell death. The graph however shows a reduction by ~30%.

4) Fig. 1e: Please show that cells remain viable at all concentrations if OxPAPC. We have in the past seen that high concentrations are toxic.

5) Fig. 1: Please provide evidence that upon transfection of LPS+OxPAPC with DOTAP, both LPS and OxPAPC reach the cytosol of cells. Furthermore, also provide evidence that when treating cells with OxPAPC without transfection reagent, OxPAPC indeed reaches the cytosol. The latter point appears important to me as it could be possible that OxPAPC treatment could block the delivery of LPS to the cytosol.

6) I agree with reviewer #3 in that it is awkward to jump between BMDMs and PMs. In fig. 1 for example the authors go from BMDMs to PM, where they test the TLR3 phenotype, polyIC priming and show that TF is not required to deliver OxPAPC. They do not show that this is indeed the case in BMDMs as well, but then perform most other experiments with poly-IC primed BMDMs. I would prefer it if they would stick to one cell type for the main figures and show their PM data in the supplementary information.

7) Fig. 1g: Could TLR4 promote OxPAPC uptake in absence of DOTAP? Please show the dependency on TLR4 if no DOTAP is used.

8) Fig. 2: The reduced level of II-1b in TLR4 Kos are most likely a result of reduced priming and thus lower pro-II-1b levels. Please provide the WB controls for these panels.

9) Fig. 5: Please provide evidence that OxPAPC binds to caspase-11 when transfected into cells or just added to the cell culture medium. The way the experiments are performed here is very artificial as the authors prepare cell lysates first, spike in high concentration OxPAPC or LPS, before performing the pull-downs.

Reviewer #4 (Remarks to the Author):

oxPAPC is created by the oxidation of 1-palmitoyl-2-arachidonyl-sn-glycero-3-phosphorylcholine (PAPC), which results in a mixture of multiple different chemical formulas including fragmented phospholipids (POVPC, PGPC, lyso-PC) and oxygenated products (PECPC and PEIPC). Though the physiological role of oxPAPC is still unclear, chemically synthesized oxPAPC allegedly has pleiotropic biological activity, including TLR2/4 antagonistic function. In another experimental setting, oxPAPC itself can be a trigger for cytokine productions through a TLR-independent mechanism. For example, a recent report (Zanoni et al., Science 2016) showed that adding oxPAPC into dendritic cells culture resulted in IL-1b release through direct activation of caspase-11. In the submitted manuscript, Chu and colleagues demonstrated that treatment of macrophages with chemically synthesized oxPAPC could attenuate caspase-11 activation in response to LPS. Mechanistically, oxPAPC was proposed to compete with LPS for caspase-11 binding. Their hypothesis is of some interest, but contradicts Zanoni et al's observations. The discrepancies are not addressed here. Furthermore, the physiological role of oxPAPC remains to be elusive and there are serious concerns about their potentially artificial experimental settings and controls used. Those aforementioned weaknesses significantly undermine the value of the submitted manuscript. In addition, the publication contains poor quality data and its unsubstantiated and controversial findings will bring needless complexity to the field. Hence I cannot recommend this for publication in Nature Communications.

Major points:

Although Chu et al showed potential antagonistic activity of chemically synthesized oxPAPC, its physiological role was not examined at all. In addition, their hypothesis contradicts, but does not directly refute, Zanoni et al's conclusion that oxPAPC can be an agonist of caspase-11. At best, Chu et al's findings will add needless complexity which requires highly context-dependent experimental setting to reproduce the phenomenon.

Chu et al refused to revisit Zanoni et al's dendritic cells study so I cannot be sure if the discrepancy can be explained by using a different type of cell. Alternatively, varying chemical formulas of oxPAPC used could be a cause of the discrepancy. Different batches of chemically synthesized oxPAPC likely contains different compositions of chemical formulas including lyso-PC, POVPC, and PEIPC. The authors failed to identify one chemical formula of oxPAPC, complicating their story and rendering it context-dependent.

Many researchers are aware of the toxicity of high dose oxPAPC. Most of the figures are missing Western blots showing pro-caspase-11, pro-II-1b, and caspase-1. Hence I am still concerned that oxPAPC promiscuously attenuates LPS response by making cells sick. Though the authors claimed that non-oxidized PAPC was used as a control, only some experiments utilized PAPC control. Important key experiments including TLR4 KO sepsis (Fig. 7 h and i), caspase-11 binding (Fig. 5 and 6), and LPS electroporation (Suppl Fig. 1f, g) still lack essential PAPC control.

PAPC control showed obvious inhibitory trend (Fig. 1b, c). Dose curve, which covers high to low range, should be shown to compare PAPC and oxPAPC. This dose curve study should be tested in TLR4 KO with both LPS electroporation and transfection conditions.

Though the authors claimed that artificial DOTAP transfection was not needed to see the oxPAPC antagonism against caspase-11 (Fig. 3c, d, Suppl Fig. 1f, g), most of data still relies on the DOTAP transfection condition, which significantly discounts the value of their findings. oxPAPC likely competes with LPS for DOTAP binding, resulting in reduction of LPS uptake. Furthermore, it is known that DOTAP itself can activate NLRP3 inflammasome.

The author claimed that oxPAPC can directly bind to caspase-11 and compete with LPS for the caspase-11 binding. However, the authors failed to compare LPS, PAPC, and oxPAPC caspase-11 binding using a quantitative method such as surface plasmon resonance. The authors also failed to examine the antagonistic efficacy of oxPAPC in a cell free system with recombinant caspase-11 and a zVAD-AMC hydrolysis assay. Hence their mechanistic hypothesis is not convincing.

Many researchers are already aware of the problem with using Hagar et al's low dose LPS model with OPTI-MEM. Insulin in OPTI-MEM artificially increases mouse susceptibility to endotoxemia. Hence the value of artificial low dose LPS model (Fig. 7 a-f) is limited. Though the authors also used a LPS high dose model, the injection of ridiculously high dose (250 mg/kg) of oxPAPC only slightly improved the survival of TLR4 KO (Fig. 7h). Hence the protective affect of oxPAPC in the model is questionable. Again, this key experiment is missing PAPC control.

Reviewer #1:

The authors present a much-improved version of their manuscript on OxPAPC-mediated inhibition of caspase-11 and caspase-4. They have addressed most of my previous concerns/question, except the issue of addressing the discrepancy between their results and the publication by the Kagan lab (Zanoni et al.) (see point 1). The authors also add a wealth of new data, which raise some new questions (see below) that should however be easily addressed before publication.

1) The authors argue that both manuscript focus on different aspects of immunity, i.e. Zanoni on adaptive immunity and DCs, and their manuscript on macrophages and sepsis. Although this is partially correct, it is obvious that the real focus of both manuscripts is the effect of OxPAPC on the non-canonical inflammasome. Here the messages are diametrically opposed, and I fear that this will be a major source of cause of confusion once the current manuscript is published.

The reason for these conflicting results might lie in the concentrations used (high vs. low concentration), the cell types (DCs vs. macrophages) and the source of the OxPAPC. Although both studies use the same source of OXPAPC, and the authors now include yet OxPAPC from other manufacturers, I am worried if batch-to-batch differences account for the conflicting results. If this would be the case, it would be a serious problem for all studies with OxPAPC since such context dependent results are hard to repeat and even harder to interpret.

To address this point, I would therefore again ask the authors to test their OxPAPC in BMDCs, using concentrations in the range used by the Kagan paper and determine if in DCs OxPAPC indeed works as an activator of the caspase-11 pathway.

We agree with reviewer 1 that analysis of DCs is important to provide a balanced view and investigated DCs alongside macrophages. We believe that our new results will complement the findings by Zanoni et al and that both manuscript together provide a better understanding of the cell type specific mechanisms of oxPAPC that are most likely derived from the vast difference in oxPAPC uptake between macrophages and DCs. Hence, oxPAPC responses seem to be cell type and concentration specific.

Macrophages: We demonstrate that oxPAPC, but not PAPC inhibits LPS-induced Caspase-11 activation, pyroptosis and IL-1 β release in macrophages. We observe this response at any tested concentration and with four different batches of oxPAPC. oxPAPC alone did not promote Caspase-11 activation and subsequent IL-1 β release in macrophages at any concentration tested, even using the very high concentrations used by Zanoni et al.

DCs: Low concentrations of oxPAPC that already completely block LPS-induced responses in macrophages, show no effect in DCs. However, very high concentrations (at least 100+ times higher than what blocks Caspase-11 in macrophages), promote a modest release of IL-1 β in DCs (3-4-fold increase), when compared to classical canonical inflammasome activators, such as nigericin (15-fold increase). Also, DCs appear to be more sensitive to LPS, as LPS priming alone causes weak IL-1 β release, which is not observed in macrophages and similar to NLRP3 activators, oxPAPC alone does not promote Caspase-11 activation independent of LPS priming.

Importantly, but contrary to the findings by Zanoni et al, we now conclusively demonstrate that this weak agonistic oxPAPC activity in DCs is completely independent of Caspase-11, but requires the canonical inflammasome, and also requires TLR4, when LPS is used to prime DCs. *These results are included as Figure 2g-j and Supplemental Figure 3.*

What is the reason for the different response? Although we do now have a concrete explanation for the difference in Caspase-11 dependent/independent LPS responses we strongly believe in our results for the following reasons: For BMDC differentiation we used purified recombinant GM-CSF, while Zanoni et al used crude GM-CSF in the form of B16 cell supernatants, which provides an inconsistent source of GM-CSF and a heterogeneous cell population (Lutz 1999, Daro 2002, Xu 2007, Kingston 2009, Guilliams 2015, Helft 2015, Lutz 2016).. In addition, we achieved very high cell purity and show that the purity of our GM-CSF differentiated DCs is 87.4% and the purity of our FLT3L/GM-CSF differentiated DCs is 96.8%, while Zanoni et al did not provide any data for the purity of their DC preparation. Our results are consistent in two different established differentiation protocols for DCs, and reveal that macrophages and DCs respond differently to oxPAPC.

Also, we determined that a main reason for the differences in the response of macrophages and DCs is based on vastly different (>10 fold) ability of these cells to take up oxPAPC (*Fig. 2i*). Using flow cytometry, we show that macrophages are much more efficient in taking up oxPAPC than DCs. Since oxPAPC does not easily reach the cytosol of DCs, it cannot block the cytosolic Caspase-11 response and consequently, displays a primarily TLR4-mediated cell surface response. Also, less LPS enters the cytosol to activate Caspase-11 and therefore also 50-fold more LPS is required in DCs to achieve a comparable IL-1 β release than in macrophages (*Fig. 2j: 5 ug LPS for DCs, as done in the Zanoni manuscript and in our DC experiments; all our other experiments with macrophages use 50 ng LPS*). In contrast, since oxPAPC enters efficiently the macrophage cytosol, it can target Caspase-11. This inability of DCs to take up oxPAPC is best emphasized by our finding that oxPAPC forced into the cytosol of DCs by liposomal delivery, redirects the TLR4 response to a Caspase-11 mediated IL-1 β release, as we observed for macrophages, where oxPAPC now blocks LPS-induced Caspase-11 activation, suggesting that the response is dictated by the availability of oxPAPC and the inability of DCs to deliver oxPAPC to the cytosol is responsible for the difference in the DC vs macrophage response.

We also find that some experiments in the Zanoni paper are inconsistent with a Caspase-11-mediated response. For example, Fig. 3F in the Zanoni manuscript shows that while LPS transfection requires Caspase-11 catalytic activity to promote canonical inflammasome-mediated IL-1 β release, but oxPAPC-induced IL-1 β release does not require this catalytic activity of Caspase-11. However, earlier studies convincingly demonstrate that Caspase-11-mediated activation of the canonical NLRP3 inflammasome for IL-1 β release absolutely requires the catalytic activity of Caspase-11, which is necessary for Pannexin-1 cleavage to induce K⁺ efflux and subsequent NLRP3 activation (Yang 2015). Furthermore,

figures showing a Caspase-11-dependent IL-1 β release show tremendously large error bars (see for example Fig. 3A).

Another limitation is that these high oxPAPC concentrations that show this weak agonistic activity in DCs, are not even found in inflamed serum (Frey 2000, Podrez 2007, Oskolkova 2010). This appears to be very reminiscent to earlier studies on TLR4, where the literature reflected an activating role of oxPAPC on TLR4 signaling, until now a consensus exists within the community that oxPAPC has inhibitory activity, unless excessive, non-physiological concentrations are used in experiments, which display an activating activity. Actually, not discussed in the Zanoni et al paper is that their finding actually conflicts with an earlier study demonstrated that low oxPAPC concentrations actually inhibit DC activation, upregulation of co-stimulatory molecules and cytokine release (Bluml 2005). Therefore, there is currently some confusion and doubt within the scientific community for the proposed Caspase-11 activating function of oxPAPC (indicated by several colleagues at meetings and even in a recent paper review), but no published report so far addressed these inconsistencies. Hence, it is our sincerest hope and intention that our manuscript will clarify and lead to a better understanding of the oxPAPC function on Caspase-11 rather than cause any more confusion than there already is.

2) The reason for these conflicting results might lie in the concentrations used (high vs. low concentration), the cell types (DCs vs. macrophages) and the source of the OxPAPC.

We believe to have compellingly ruled out the source of oxPAPC as the cause for our results, by using several different batches of oxPAPC, including the same source used in the Zanoni paper. All batches show identical responses in macrophages and DCs. *This is shown in Supplementary Figure 1f-h.*

Also, the concentration of oxPAPC does not account for the differences. All tested concentrations are inhibitory in macrophages and do not show any agonistic activity. Low, physiological oxPAPC concentrations show no activity in DCs, while excessive levels of oxPAPC that are not be present in inflamed serum, show very weak agonistic activity only in DCs. Of note is that analysis of IL1- β at >16h, as done in the Zanoni paper, already causes weak release of IL-1 β in DCs, but not macrophages. Importantly, this weak agonistic activity is completely independent of Caspase-11.

Collectively, our results show that there is a cell type-specific response to oxPAPC in macrophages and DCs.

2) Fig. 1b: Please provide the p-values for the difference between P3C4/LPS/DOTAP and P3C4/LPS/DOTAP+PAPC. Is this really not a significant reduction?

In some experiments, we observed minor inhibitory activity of PAPC, which, however, was always much less pronounced compared to oxPAPC, and usually did not reach statistical significance, but clearly represents a trend. We suspect that the experimental conditions are responsible for this effect and that it is based on some level of spontaneous oxidation of PAPC when PAPC was exposed to oxygen during incubations. Using a fresh aliquot usually showed less effect, which got more pronounced with time. In fact, you achieve full oxidation by exposure to air for >24h, which is a routine protocol to oxidize PAPC. We purchase PAPC at larger quantities (5 mg) and aliquot under nitrogen for further use, but even this procedure exposes PAPC briefly to oxygen. In this particular experiment, the p value is 0.1 and was therefore not marked as significant.

3) Fig. 1d: In the text the authors claim that Wt and TLR3 KOs show ‘comparable’ level of cell death. The graph however shows a reduction by ~30%.

The reviewer is correct. We noted this reduced activity in *Tlr3*^{-/-} (when primed with LPS) and also in *Tlr4*^{-/-} mice (when primed with Pam3CSK4 or poly(I:C)). We did not make a specific point of this, as this is not the focus of this study and experiments are in progress to determine the mechanism of this reduced activity. We notice the same phenomena in BMDMs (*Figure 1d*) and peritoneal macrophages (*Supplementary Figure 1b, which is the previous Figure 1d*). We mention this effect in the revised text and discussion. We show that even liposomal delivery of LPS engages TLR4 and causes a transcriptional effect, which we now include as *Supplementary Figure 2*. We can only speculate that this may have to do with the previously described TLR-mediated rapid inflammasome activation (Lin 2014).

4) Fig. 1e: Please show that cells remain viable at all concentrations of OxPAPC. We have in the past seen that high concentrations are toxic.

In our hands, at the concentrations we used, we did not observe oxPAPC-induced toxicity and oxPAPC alone does not cause any LDH release in any of our presented figures, which is also a readout for cell death. To the contrary, addition of even high concentrations of oxPAPC prevents LPS-induced cytotoxicity. As requested, we also performed flow cytometry to determine cell viability after oxPAPC treatment and transfection showing the concentration used in most of our experiments in macrophages, which does not at all affect viability and is comparable to mock treatment or transfection. *We added this information as Supplementary Figure 1c.*

5) Fig. 1: Please provide evidence that upon transfection of LPS+OxPAPC with DOTAP, both LPS and OxPAPC reach the cytosol of cells. Furthermore, also provide evidence that when treating cells with OxPAPC without transfection reagent, OxPAPC indeed reaches the cytosol. The latter point appears important to me as it could be possible that OxPAPC treatment could block the delivery of LPS to the cytosol.

We provide evidence that LPS reaches the cytosol upon transfection and incubation, as it causes Caspase-11 activation in the cytosol. However, as suggested, we also performed flow cytometric analysis of oxPAPC and LPS uptake.

a) We demonstrate that transfection of LPS delivers identical levels of LPS to the cytosol as co-transfection of LPS and oxPAPC (*Supplementary Figure 1j*).

b) We show that treatment of macrophages with biotinylated oxPAPC results in efficient uptake into the cytosol in a dose-dependent manner, and that macrophages are much more efficient than DCs, which require much higher oxPAPC to enter the cytosol (*Figure 2i*).

c) In addition, we demonstrate that oxPAPC can compete with LPS for Caspase-11 binding in a setting independent of LPS delivery. We demonstrate that oxPAPC not only prevents purified LPS-mediated Caspase-11 activation, but also LPS delivery by live bacteria and outer membrane vesicles (OMV), further supporting the argument that oxPAPC acts on Caspase-11.

6) I agree with reviewer #3 in that it is awkward to jump between BMDMs and PMs. In fig. 1 for example the authors go from BMDMs to PM, where they test the TLR3 phenotype, polyIC priming and show that TF is not required to deliver OxPAPC. They do not show that this is indeed the case in BMDMs as well, but then perform most other experiments with poly-IC primed BMDMs. I would prefer it if they would stick to one cell type for the main figures and show their PM data in the supplementary information.

We agree and included new results obtained with *Tlr3*^{-/-} BMDM in the main text (*Figure 1d*) and show the PM data as *Supplementary Figure 1b*.

7) Fig. 1g: Could TLR4 promote OxPAPC uptake in absence of DOTAP? Please show the dependency on TLR4 if no DOTAP is used.

We demonstrate that during *Salmonella* infection, treatment with oxPAPC, but not PAPC is able to prevent Caspase-11 activation and pyroptosis also in *Tlr4*^{-/-} mice, demonstrating that TLR4 is not required for oxPAPC uptake (*Figure 3d*). As requested, we also analyzed oxPAPC uptake by flow cytometry in WT and *Tlr4*^{-/-} BMDMs, which revealed no real differences in oxPAPC uptake (*Supplementary Figure 1e*).

8) Fig. 2: The reduced level of Il-1b in TLR4 KO are most likely a result of reduced priming and thus lower pro-Il-1b levels. Please provide the WB controls for these panels.

We show that liposomal delivery of also LPS engages TLR4 and causes a transcriptional effect, as determined by qPCR and by immunoblot, which we now include as *Supplementary Figure 2*.

9) Fig. 5: Please provide evidence that OxPAPC binds to caspase-11 when transfected into cells or just added to the cell culture medium. The way the experiments are performed here is very artificial as the authors prepare cell lysates first, spike in high concentration OxPAPC or LPS, before performing the pull-downs.

This strategy was necessary, due to the biochemical approach to detect oxPAPC/Caspase-11 binding. The miniscule amounts of oxPAPC and LPS entering cells, which are sufficient to induce functional responses, was not sufficiently sensitive to allow detection. To our knowledge, no study has so far shown binding of LPS to Caspase-11 in cells and was only done with recombinant Caspase-11. Therefore, we had to demonstrate binding in a cell free system, which we believe provides the necessary evidence that oxPAPC binds to Caspase-4 and Caspase-11.

Reviewer #4:

oxPAPC is created by the oxidation of 1-palmitoyl-2-arachidonoyl-sn-glycero-3-phosphorylcholine (PAPC), which results in a mixture of multiple different chemical formulas including fragmented phospholipids (POVPC, PGPC, lyso-PC) and oxygenated products (PECPC and PEIPC). Though the physiological role of oxPAPC is still unclear, chemically synthesized oxPAPC allegedly has pleiotropic biological activity, including TLR2/4 antagonistic function. In another experimental setting, oxPAPC itself can be a trigger for cytokine productions through a TLR-independent mechanism. For example, a recent report (Zanoni et al., Science 2016) showed that adding oxPAPC into dendritic cells culture resulted in IL-1b release through direct activation of caspase-11. In the submitted manuscript, Chu and colleagues demonstrated that treatment of macrophages with chemically synthesized oxPAPC could attenuate caspase-11 activation in response to LPS. Mechanistically, oxPAPC was proposed to compete with LPS for caspase-11 binding. Their hypothesis is of some interest, but contradicts Zanoni et al's observations. The discrepancies are not addressed here. Furthermore, the physiological role of oxPAPC remains to be elusive and there are serious concerns about their potentially artificial experimental settings and controls used. Those aforementioned weaknesses significantly undermine the value of the submitted manuscript. In addition, the publication contains poor quality data and its unsubstantiated and controversial findings will bring needless complexity to the field. Hence I cannot recommend this for publication in Nature Communications.

Our revised manuscript provides some answers to the discrepancies between our manuscript and the manuscript by Zanoni et al, which itself contradicted previously published articles (see above). We now find that DCs respond differently to oxPAPC than macrophages, due to the ineffective uptake of oxPAPC in DCs. Therefore, we believe that our study will clarify the differences between Zanoni's and our as well

as other previously published findings, and is therefore needed by the scientific community to gain a better understanding and a more clear picture of the rather complex, cell type specific response. Therefore, only both manuscripts together provide the necessary balanced and accurate view on cell type and concentration specific responses.

In addition we provide an extensive amount of controls and most results are substantiated by several methods.

a) As detailed for reviewer 1 (#1), we provide compelling evidence that macrophages and DCs respond differently to oxPAPC, caused by the inability of DCs to efficiently take up oxPAPC. This very modest agonistic activity, barely above LPS-mediated effects, is only achieved with huge oxPAPC concentrations, 100+ fold higher than what efficiently blocks Caspase-11 in macrophages.

b) We provide compelling evidence that the response in DCs is mediated by TLR4-mediated NLRP3 inflammasome activation, but does not engage Caspase-11 (see above).

Major points:

Although Chu et al showed potential antagonistic activity of chemically synthesized oxPAPC, its physiological role was not examined at all. In addition, their hypothesis contradicts, but does not directly refute, Zanoni et al's conclusion that oxPAPC can be an agonist of caspase-11. At best, Chu et al's findings will add needless complexity which requires highly context-dependent experimental setting to reproduce the phenomenon.

We now demonstrate that this response is indeed context dependent, as we demonstrate that macrophages and DCs respond different to oxPAPC. However, macrophages respond to low oxPAPC concentrations, while DCs require a very high oxPAPC concentration, which has been previously shown not to exist in serum and we demonstrate that this is completely Caspase-11 independent. Please see our response to reviewer 1. The phenomenon in DCs requires actually more non-physiological conditions, due to the required high oxPAPC concentration.

Chu et al refused to revisit Zanoni et al's dendritic cells study so I cannot be sure if the discrepancy can be explained by using a different type of cell. Alternatively, varying chemical formulas of oxPAPC used could be a cause of the discrepancy. Different batches of chemically synthesized oxPAPC likely contains different compositions of chemical formulas including lyso-PC, POVPC, and PEIPC. The authors failed to identify one chemical formula of oxPAPC, complicating their story and rendering it context-dependent.

We now included DCs in our analysis and demonstrate a cell type specific response, but did not observe a batch effect. Please see our response to reviewer #1 above. The context dependent effect is that DCs are largely unable to take up oxPAPC, explaining a TLR4, but not Caspase-11 mediated response. The manuscript by Zanoni et al actually does not even demonstrate that oxPAPC is actually taken up under their conditions, likely, because DCs only take up oxPAPC at high concentrations.

Many researchers are aware of the toxicity of high dose oxPAPC. Most of the figures are missing Western blots showing pro-caspase-11, pro-IL-1b, and caspase-1. Hence I am still concerned that oxPAPC promiscuously attenuates LPS response by making cells sick. Though the authors claimed that non-oxidized PAPC was used as a control, only some experiments utilized PAPC control. Important key experiments including TLR4 KO sepsis (Fig. 7 h and i), caspase-11 binding (Fig. 5 and 6), and LPS electroporation (Suppl Fig. 1f, g) still lack essential PAPC control.

In our hands, at the concentrations we use, we did not observe oxPAPC-induced toxicity and oxPAPC alone does not cause any LDH release. We include now cell viability assay results demonstrating that the

oxPAPC concentrations used in our study does not at all affect viability. If at all, we would expect that the much higher concentrations used in the study by Zanoni et al would cause toxicity and canonical inflammasome activation following the release of endogenous DAMPs. Furthermore, PAPC controls in key experiments demonstrate no effect on LDH release, cytotoxicity and survival. In particular, oxPAPC does not increase LPS-induced toxicity, but rather reduces toxicity, further eliminating the argument of a toxicity-caused effect. PAPC controls were repeated for all key experiments and included in Figures: 1b, c, m; 2d, e, f; 3c, d; 4c, d, g, h, i; 7b, g, h, i; Supplemental Fig 1d, i,

PAPC control showed obvious inhibitory trend (Fig. 1b, c). Dose curve, which covers high to low range, should be shown to compare PAPC and oxPAPC. This dose curve study should be tested in TLR4 KO with both LPS electroporation and transfection conditions.

PAPC is kept under nitrogen, but is oxidized to some extent upon exposure to air during experiments, which we believe accounts for these minor effects, which nevertheless do not reach statistical significance. We use the identical concentration of oxPAPC and PAPC as controls in key experiments and since the higher doses of PAPC do not show any effect we do not see the reason for performing this with even lower concentrations. Similarly, we do not understand the reason to include higher concentrations of PAPC than we use for oxPAPC in our experiments.

Though the authors claimed that artificial DOTAP transfection was not needed to see the oxPAPC antagonism against caspase-11 (Fig. 3c, d, Suppl Fig. 1f, g), most of data still relies on the DOTAP transfection condition, which significantly discounts the value of their findings. oxPAPC likely competes with LPS for DOTAP binding, resulting in reduction of LPS uptake. Furthermore, it is known that DOTAP itself can activate NLRP3 inflammasome.

We actually do not observe any DOTAP-mediated inflammasome activation or effects on viability. We carefully optimized our approach to not cause DOTAP effects and demonstrate DOTAP controls in all experiments, demonstrating that oxPAPC does not interfere with DOTAP. DOTAP in our experiments also does not cause any release of any of the tested cytokines (*Fig. 2a-f*), clearly eliminating this concern. Furthermore, and most importantly, we also use transfection-independent approaches, as well as delivery of OMV and infection with Salmonella. Most experiments utilize transfections on purpose as a mean to largely bypass TLR4, which is absolutely necessary for the conclusion and not possible by in addition having a TLR4 response that occurs by LPS treatment, which is well established to be modulated by oxPAPC. We now also show that oxPAPC does not alter the uptake of LPS (*Supplementary Figure 1j*).

The author claimed that oxPAPC can directly bind to caspase-11 and compete with LPS for the caspase-11 binding. However, the authors failed to compare LPS, PAPC, and oxPAPC caspase-11 binding using a quantitative method such as surface plasmon resonance. The authors also failed to examine the antagonistic efficacy of oxPAPC in a cell free system with recombinant caspase-11 and a zVAD-AMC hydrolysis assay. Hence their mechanistic hypothesis is not convincing.

We opted for a functional and cell based approach rather than one relying on recombinant proteins, which in our view is much more prone for artificial results. We never made the point or intended to indicate any binding affinities, but rather base our conclusion on functional assays. We did not feel the necessity to use the artificial zVAD cleavage assay, as we demonstrate the oxPAPC effect on the much more relevant endogenous substrates for Caspase-11 in cell based assays, which in our opinion are much more relevant, convincing and compelling.

Many researchers are already aware of the problem with using Hagar et al's low dose LPS model with OPTI-MEM. Insulin in OPTI-MEM artificially increases mouse susceptibility to endotoxemia. Hence the value of artificial low dose LPS model (Fig. 7 a-f) is limited. Though the authors also used a LPS high dose model, the injection of ridiculously high dose (250 mg/kg) of oxPAPC only slightly improved the survival of TLR4 KO (Fig. 7h). Hence the protective effect of oxPAPC in the model is questionable. Again, this key experiment is missing PAPC control.

We are aware of this “problem”. We do not agree that the value of the model where we “enhance” the LPS response with Opti-MEM (low dose insulin) is of low value. In fact, sepsis affects glucose metabolism and glycemic control is part of any acute sepsis treatment in the ICU. This model has been vetted and allowed us to use low LPS doses and consequently low oxPAPC concentrations as a means to demonstrate this oxPAPC effect. We never disputed or hid Opti-MEM has been used and are aware that Opti-MEM sensitizes mice to LPS, which is the intended purpose and the reason why we opted for this model. Importantly, this combination of LPS and Opti-MEM still requires LPS-mediated Caspase-11 activation, which was our intention and allowed us to demonstrate the oxPAPC effect and that PAPC has no effect on survival.

The reviewer refers to the context dependent mechanism studied by Zanoni et al. However, this phenomenon only becomes effective, when cells or mice are also treated or injected with LPS, while oxPAPC had no effect in vitro or in vivo on its own, in spite of being linked to also transcriptional responses. In fact, they have to use very high LPS concentrations ($1 \mu\text{g mL}^{-1}$ in vitro) to trigger any weak response. In particular, in vivo LPS injection at much lower concentrations than used in their study ($7 \mu\text{g}$) already causes potent inflammatory responses even in the absence of an adjuvant in our hands that is partially dependent on Caspase-11 (not shown), and Zanoni et al LPS is emulsified in an adjuvant, which is similar to the purpose of the Opti-MEM used in our low dose LPS study and loss of Caspase-11 may affect the LPS-mediated part of the response.

The high dose LPS response model was requested by reviewers. The problem with the high dose LPS experiment, as the reviewer indicated, is the need to compensate with oxPAPC for this high amount of LPS, which was the rationale for using the low LPS model in the first place. We did not perform a dose response in vivo and opted for a dose to achieve strong competition for LPS. However, the reviewer does not mention that a much lower dose was equally effective in the low dose LPS model, where we injected only $10 \mu\text{g}/\text{mouse}$, *which is 1/7th the concentration used by Zanoni et al.* However, the reviewer overlooked that even though we only rescue 25% of mice, this is the same protection conferred by loss of Caspase-11. In our hands, this is the best Caspase-11 protection we can obtain with Caspase-11 KO mice. Hence, we cannot expect a better survival than in Caspase-11 KO mice. Importantly, this effect is independent of TLR4. While lower LPS concentrations are closer to the 60% survival of Caspase-11 KO mice reported earlier, 100% of TLR4 KO mice survive, constraining us to use this “compromise”. We now repeated the experiment with PAPC as a control, again showing no protective effect of PAPC also in the high dose LPS model.

We realize the defined conditions of our in vivo experiments, which are necessary to exclude TLR4 effects and enable and visualize Caspase-11-mediated effects. However, we strongly believe that the combination of both LPS models provide the evidence that oxPAPC can protect from LPS in a Caspase-11-dependent manner. Furthermore, acute LPS-induced shock is partially prevented by oxPAPC, either indicating that this response is mostly dependent on macrophages or that the physiological response is inhibiting Caspase-11 rather than activating it.

Overall, we feel that we provide compelling evidence that oxPAPC elicits a cell type specific response in macrophages compared to DCs and that the diminished uptake of oxPAPC in DCs causes a predominantly TLR4 driven cell surface response rather than acting on the intracellular Caspase-11. However, intracellular oxPAPC inhibits intracellular LPS driven Caspase-11 activation in DCs and macrophages.

References

- Bluml, S., S. Kirchberger, V. N. Bochkov, G. Kronke, K. Stuhlmeier, O. Majdic, G. J. Zlabinger, W. Knapp, B. R. Binder, J. Stockl and N. Leitinger (2005). "Oxidized phospholipids negatively regulate dendritic cell maturation induced by TLRs and CD40." *J Immunol* **175**(1): 501-508.
- Daro, E., E. Butz, J. Smith, M. Teepe, C. R. Maliszewski and H. J. McKenna (2002). "Comparison of the functional properties of murine dendritic cells generated in vivo with Flt3 ligand, GM-CSF and Flt3 ligand plus GM-SCF." *Cytokine* **17**(3): 119-130.
- Frey, B., R. Haupt, S. Alms, G. Holzmann, T. Konig, H. Kern, W. Kox, B. Rustow and M. Schlame (2000). "Increase in fragmented phosphatidylcholine in blood plasma by oxidative stress." *J Lipid Res* **41**(7): 1145-1153.
- Guilliams, M. and B. Malissen (2015). "A Death Notice for In-Vitro-Generated GM-CSF Dendritic Cells?" *Immunity* **42**(6): 988-990.
- Helft, J., J. Bottcher, P. Chakravarty, S. Zelenay, J. Huotari, B. U. Schraml, D. Goubau and C. Reis e Sousa (2015). "GM-CSF Mouse Bone Marrow Cultures Comprise a Heterogeneous Population of CD11c(+)MHCII(+) Macrophages and Dendritic Cells." *Immunity* **42**(6): 1197-1211.
- Kingston, D., M. A. Schmid, N. Onai, A. Obata-Onai, D. Baumjohann and M. G. Manz (2009). "The concerted action of GM-CSF and Flt3-ligand on in vivo dendritic cell homeostasis." *Blood* **114**(4): 835-843.
- Lin, K. M., W. Hu, T. D. Troutman, M. Jennings, T. Brewer, X. Li, S. Nanda, P. Cohen, J. A. Thomas and C. Pasare (2014). "IRAK-1 bypasses priming and directly links TLRs to rapid NLRP3 inflammasome activation." *Proc Natl Acad Sci U S A* **111**(2): 775-780.
- Lutz, M. B., K. Inaba, G. Schuler and N. Romani (2016). "Still Alive and Kicking: In-Vitro-Generated GM-CSF Dendritic Cells!" *Immunity* **44**(1): 1-2.
- Lutz, M. B., N. Kukutsch, A. L. Ogilvie, S. Rossner, F. Koch, N. Romani and G. Schuler (1999). "An advanced culture method for generating large quantities of highly pure dendritic cells from mouse bone marrow." *J Immunol Methods* **223**(1): 77-92.
- Oskolkova, O. V., T. Afonyushkin, B. Preinerstorfer, W. Bicker, E. von Schlieffen, E. Hainzl, S. Demyanets, G. Schabbauer, W. Lindner, A. D. Tselepis, J. Wojta, B. R. Binder and V. N. Bochkov (2010). "Oxidized phospholipids are more potent antagonists of lipopolysaccharide than inducers of inflammation." *J Immunol* **185**(12): 7706-7712.
- Podrez, E. A., T. V. Byzova, M. Febbraio, R. G. Salomon, Y. Ma, M. Valiyaveetil, E. Poliakov, M. Sun, P. J. Finton, B. R. Curtis, J. Chen, R. Zhang, R. L. Silverstein and S. L. Hazen (2007). "Platelet CD36 links hyperlipidemia, oxidant stress and a prothrombotic phenotype." *Nat Med* **13**(9): 1086-1095.
- Xu, Y. K., Y. F. Zhan, A. M. Lew, S. H. Naik and M. H. Kershaw (2007). "Differential development of murine dendritic cells by GM-CSF versus flt3 ligand has implications for inflammation and trafficking." *Journal of Immunology* **179**(11): 7577-7584.
- Yang, D., Y. He, R. Munoz-Planillo, Q. Liu and G. Nunez (2015). "Caspase-11 Requires the Pannexin-1 Channel and the Purinergic P2X7 Pore to Mediate Pyroptosis and Endotoxic Shock." *Immunity* **43**(5): 923-932.
- Zanoni, I., Y. Tan, M. Di Gioia, A. Broggi, J. Ruan, J. Shi, C. A. Donado, F. Shao, H. Wu, J. R. Springstead and J. C. Kagan (2016). "An endogenous caspase-11 ligand elicits interleukin-1 release from living dendritic cells." *Science* **352**(6290): 1232-1236.

Reviewers' comments:

Reviewer #1 (Remarks to the Author):

The authors have addressed my criticisms. I have only few small points that should be addressed before publication:

- 1) Statistics: From the figure legends it is impossible to deduce what the error bars really show. Most data are representative of 3 or more experiments, but do the error bars in bar graphs show a mean of these 3 experiments, or the average of triplicate/quadruplicate wells? In addition, I think that showing the s.e.m is not the correct analysis, and that s.d. needs to be shown.
- 2) Fig. 1I: Here it would be ok to show the full GSDMD blot. The blot for p30 is currently quite awkwardly cut, and shows part of a crossreactive band.
- 3) Figure 4L, lane 1: What is the band in the polyIC primed cells that has a size similar to p30?
- 4) Figure 2g: LDH data and immunoblots for processed vs. mature IL-1b in SNs are necessary. It is possible that the very low levels of IL-1b release detected after OxPAPC treatment of BMDCs are the result of lysis, and that it is not processed IL-1b, but full-length pro-IL-1b that is detected by ELISA in the SNs.
- 5) Figure 2g-h: Please examine the labels: Is it really ug ml-2?
- 6) Figure 2g: How would the IL-1b look if DOTAP is used to transfect OxPAPC?
- 7) In the text (line 256) the authors suggest that OxPAPC activates an NLRP3 inflammasome in BMDCs. There is however no evidence for this, except that it is a TLR4, ASC and Casp-1-dependent response. Please examine NLRP3 BMDCs to confirm the dependency on these receptors.
- 8) Text line 133: 30% reduction in TLR3 KO cannot be called a "slight" reduction
- 9) Text lines 535-536: This sentence is confusing, as the authors do suggest that OxPAPC has no inhibitory activity in BMDCs on the non-canonical pathway triggered by LPS. However Figure 2j, shows that OxPAPC inhibits IL-1b production elicited by DOTAP-LPS treatment in BMDCs.
- 10) The authors are certainly aware of the new study by the Kagan lab published recently in *Immunity* (Zanoni et al. 2017). This study suggests that in contrast to the conclusion reached here, the differential response between BMDMs and BMDCs is not only linked to a "cell type specific response", but is also dependent on the composition of OxPAPC. It appears that some components of OxPAPC act anti-inflammatory, while others act pro-inflammatory (POVPC, PGPC), and that different cell types display differential sensitivities to these compounds. While it would have been interesting to see how POVPC and PGPC behave in the authors' hands, and if this response is caspase-11 dependent (as suggested by Kagan and co.), I think that it should be sufficient to include a discussion of Zanoni et al. *Immunity* 2017 in relation to the authors' own data.

Reviewer #4 (Remarks to the Author):

Chu et al revised well with additional data and answered to all my concerns. With a minor revision, the manuscript is deserved for the publication.

Minor point

The problem in using the artificial low dose LPS model (with OPTI-MEM medium) was published. This should be cited to explain the model used in the manuscript.

Lipopolysaccharide Potentiates Insulin-Driven Hypoglycemic Shock.

Hagar JA, Edin ML, Lih FB, Thurlow LR, Koller BH, Cairns BA, Zeldin DC, Miao EA.

J Immunol. 2017 Oct 16. pii: ji1700820. doi: 10.4049/jimmunol.1700820.

Reviewer Response

We thank both reviewers for their time and reviewer #1 for bringing additional constructive critique to further improve this manuscript. We carefully addressed the remaining critique points as summarized below and highlighted all changes in the text in yellow:

Reviewer #1:

The authors have addressed my criticisms. I have only few small points that should be addressed before publication:

1) Statistics: From the figure legends it is impossible to deduce what the error bars really show. Most data are representative of 3 or more experiments, but do the error bars in bar graphs show a mean of these 3 experiments, or the average of triplicate/quadruplicate wells?

In addition, I think that showing the s.e.m is not the correct analysis, and that s.d. needs to be shown.

We indicated in the statistical methods section and each figure legend that each graph represents one representative experiment of three or more repeats, each consisting of at least triplicate samples. We agree with the reviewer that s.e.m. is not the best presentation and we changed all graphs to show the error bars as s.d. in this revised manuscript.

2) Fig. 1I: Here it would be ok to show the full GSDMD blot. The blot for p30 is currently quite awkwardly cut, and shows part of a crossreactive band.

We show a larger area from the blot in the revised manuscript. The band right above the p30 is an artifact from these blots, as it is also present in empty lanes and the marker lane. We marked this cross-reactive band with an asterisk.

3) Figure 4L, lane 1: What is the band in the polyI:C primed cells that has a size similar to p30?

In primary human macrophages, we only obtained very inefficient cleavage of GSDMD and in addition, available anti-human GSDMD antibodies also yield cross-reactive bands, as the one above the p30, but based on size and controls, we are confident in these results (now Fig. 5I) . We also included full blots in the supplement.

4) Figure 2g: LDH data and immunoblots for processed vs. mature IL-1 β in SNs are necessary. It is possible that the very low levels of IL-1 β release detected after OxPAPC treatment of BMDCs are the result of lysis, and that it is not processed IL-1 β , but full-length pro-IL-1 β that is detected by ELISA in the SNs.

These cells were not transfected with LPS, but only primed with extracellular LPS, which does not cause increased cell death under our conditions in macrophages or DCs. We included LDH release as a readout for pyroptosis for these conditions in Supplementary Figure 4b and c. Although DCs showed a little more basal LDH release than BMDMs, neither LPS, oxPAPC or LPS/oxPAPC treatment further increased LDH release. Importantly, this basal LDH release was independent of TLR4, the canonical and the non-canonical inflammasome. We added this information as Supplementary Figure 4b, c. Based on these result, the released IL-1 β quantified by ELISA was not caused by cytotoxicity. To further back up this finding, we performed immunoblot of total cell lysates and culture SN, which also demonstrated that

mature IL-1 β is released. Conditions that cause IL-1 β processing and release, also release some pro-IL-1 β , which is shown in numerous studies and known in the field. Since the pro-IL-1 β antibody has low affinity for the processed IL-1 β , we also included results obtained with a cleavage-specific mature IL-1 β antibody. These results further underscore that IL-1 β processing and release in response to LPS/oxPAPC is completely independent of Caspase-11, partially dependent on the canonical inflammasome, and fully dependent on TLR4. These results are consistent with our ELISA results and we show these results in Supplementary Figure 4d.

5) Figure 2g-h: Please examine the labels: Is it really ug ml-2?

We corrected this mistake, which should be $\mu\text{g ml}^{-1}$ (now Fig. 3a, b).

6) Figure 2g: How would the IL-1b look if DOTAP is used to transfect OxPAPC?

We show this result in Figure 2j (now Figure 3e) and discussed this in the manuscript. Briefly, as we show in our previous revision, transfection of oxPAPC into DCs results in complete inhibition of IL-1 β release. While this pathway depends on the delivery of LPS to the cytosol and on Caspase-11, merely priming DCs (as well as macrophages) with LPS is not sufficient to cause Caspase-11-dependent cell death or IL-1 β release. However, prolonged LPS treatment signals through a TLR4 and canonical inflammasome dependent, but Caspase-11 independent pathway that leads to IL-1 β production in DCs, which is not inhibited by oxPAPC (see Figure 3a and b and Supplementary Figure 4f).

7) In the text (line 256) the authors suggest that oxPAPC activates an NLRP3 inflammasome in BMDCs. There is however no evidence for this, except that it is a TLR4, ASC and Casp-1-dependent response. Please examine NLRP3 BMDCs to confirm the dependency on this receptor.

We based this statement on results shown by Zanoni et al who indicated NLRP3 inflammasome activation by oxPAPC. We performed our own experiments with *Nlrp3*^{-/-} BMDC alongside WT and *Casp11*^{-/-} BMDC for this revised manuscript, which confirmed that oxPAPC triggers NLRP3 dependent but Caspase-11 independent IL-1 β release in BMDC (Figure 3b).

8) Text line 133: 30% reduction in TLR3 Kos cannot be called a “slight” reduction

We removed this statement.

9) Txt lines 535-536: This sentence is confusing, as the authors do suggest that OxPAPC has no inhibitory activity in BMDCs on the non-canonical pathway triggered by LPS. However, Figure 2j, shows that OxPAPC inhibits IL-1b production elicited by DOTAP-LPS treatment in BMDCs.

We only achieved this effect now shown in Fig. 3e, when we transfect oxPAPC and LPS. This is in contrast to macrophages, which take up oxPAPC in sufficient quantities to target Caspase-11. In our hands, treatment with oxPAPC in the absence of DOTAP does not affect Caspase-11 in DCs and therefore, in contrast to macrophages, transfection of oxPAPC into DCs is therefore artificial. We show by flow cytometry that DCs take up oxPAPC much less efficient compared to macrophages, which we predict is the reason that not sufficient oxPAPC can be taken up by DCs to target Caspase-11 and therefore preferentially target cell surface receptors. We clarified this statement in the text.

10) The authors are certainly aware of the new study by the Kagan lab published recently in immunity (Zanoni et al. 2017). This study suggests that in contrast to the conclusion reached here, the differential response between BMDMs and BMDCs is not only linked to a ‘cell type specific response’, but is also dependent on the composition of OxPAPC. It appears that some components of OxPAPC act anti-inflammatory, while other act pro-inflammatory (POVPC, PGPC), and that different cell types display differential sensitivities to these compounds.

While it would have been interesting to see how POVPC and PGPC behave in the authors hands, and if this response is caspase-11 dependent (as suggested by Kagan and co.), I think that it should be sufficient to include a discussion of Zanoni et al. Immunity 2017 in relation to the authors own data.

This cell type specific sensitivity is a likely explanation and we included this in the discussion of this revised manuscript. However, our data do not support that select oxidation products are pro-inflammatory, while others are anti-inflammatory. As suggested, we tested individual oxPAPC components and two of the major products, POVPC and PGPC, but not DMPC, inhibit Caspase-11 in macrophages. We added this result as Figure 1n. POPVP and PGPC activated DCs in the study by Zanoni et al., while DMPC also did not show activity. Therefore, the same oxidation products that are pro-inflammatory in DCs are anti-inflammatory in macrophages. A pro-inflammatory response of these phospholipids is only supported in the literature, when excessive concentrations and prolonged incubation times are used to activate macrophages, which we believe may account for the differences in our studies. Such high oxPAPC concentrations have been shown to be not physiological. In fact, PGPC and POVPC represent less than 10% of oxPAPC, hence to promote this response in macrophages requires even >10-fold more than the already non-physiological concentration used in these studies to trigger the weak response in DCs. Published studies using physiological concentrations support an anti-inflammatory response of oxPAPC.

Reviewer #4:

Chu et al revised well with additional data and answered to all my concerns. With a minor revision, the manuscript is deserved for the publication.

Minor point

The problem in using the artificial low dose LPS model (with OPTI-MEM medium) was published. This should be cited to explain the minor model used in the manuscript.

Lipopolysaccharide Potentiates Insulin-Driven Hypoglycemic Shock.

Hagar JA, Edin ML, Lih FB, Thurlow LR, Koller BH, Cairns BA, Zeldin DC, Miao EA.

J Immunol. 2017 Oct 16. pii: j11700820. doi: 10.4049/jimmunol.1700820.

We included this reference in the revised manuscript, which was published after our previous re-submission as reference 57.

REVIEWERS' COMMENTS:

Reviewer #1 (Remarks to the Author):

The authors have fully addressed my comments, and I support the publication of the study in Nature Communications. The paper is a valuable contribution to the field and will help to trigger a much necessary discussion on the real link between inflammasomes and OxPAPC stimulation, given that many lab have failed to repeat the results published by Jon Kagan's group.